# DADAM: A CONSENSUS-BASED DISTRIBUTED ADAPTIVE GRADIENT METHOD FOR ONLINE OPTIMIZATION

## ABSTRACT

Online and stochastic optimization methods such as SGD, ADAGRAD and ADAM are key algorithms in solving large-scale machine learning problems including deep learning. A number of schemes that are based on communications of nodes with a central server have been recently proposed in the literature to parallelize them. A bottleneck of such centralized algorithms lies on the high communication cost incurred by the central node. In this paper, we present a new consensus-based distributed adaptive moment estimation method (DADAM) for online optimization over a decentralized network that enables data parallelization, as well as decentralized computation. Such a framework not only can be extremely useful for learning agents with access to only local data in a communication constrained environment, but as shown in this work also outperform centralized adaptive algorithms such as ADAM for certain realistic classes of loss functions. We analyze the convergence properties of the proposed algorithm and provide a *dynamic regret* bound on the convergence rate of adaptive moment estimation methods in both stochastic and deterministic settings. Empirical results demonstrate that DADAM works well in practice and compares favorably to competing online optimization methods.

## 1 INTRODUCTION

Online optimization is a fundamental procedure for solving a wide range of machine learning problems Shalev-Shwartz et al. (2012); Hazan et al. (2016). It can be formulated as a repeated game between a learner (algorithm) and an adversary. The learner receives a streaming data sequence, sequentially selects actions, and the adversary reveals the convex or nonconvex losses to the learner. A standard performance metric for an online algorithm is *regret*, which measures the performance of the algorithm versus a static benchmark Zinkevich (2003); Hazan et al. (2016). For example, the benchmark could be an optimal point of the online average of the loss (local cost) function, had the learner known all the losses in advance. In a broad sense, if the benchmark is a fixed sequence, the regret is called *static*. Recent work on online optimization has investigated the notion of *dynamic* regret Zinkevich (2003); Hall & Willett (2015); Besbes et al. (2015). Dynamic regret can take the form of the cumulative difference between the instantaneous loss and the minimum loss. For convex functions, previous studies have shown that the dynamic regret of online gradient-based methods can be upper bounded by $O(\sqrt{TD_T})$, where $D_T$ is a measure of regularity of the comparator sequence or the function sequence Zinkevich (2003); Hall & Willett (2015). This bound can be improved to $O(D_T)$ Mokhtari et al. (2016); Zhang et al. (2017) when all the cost functions are strongly convex and smooth.

Decentralized nonlinear programming has received a great deal of interest in diverse scientific and engineering fields Tsitsiklis et al. (1986); Li et al. (2002); Rabbat & Nowak (2004); Lesser et al. (2012). The key problem involves optimizing a cost function $f(x) = \frac{1}{n}\sum_{i=1}^{n} f_i(x)$, where $x \in \mathbb{R}^p$ and each $f_i$ is only known to the individual agent $i$ in a connected network of $n$ agents. The agents collaborate by successively sharing information with other agents located in their neighborhood with the goal of jointly converging to the network-wide optimal argument Nedic & Ozdaglar (2009). Compared to optimization procedures involving a fusion center that collects data and performs the computation, decentralized nonlinear programming enjoys the advantage of scalability to network sizes, robustness to dynamic topologies, and privacy preservation in data-sensitive applications Yuan et al. (2016); Shi et al. (2015); Jiang et al. (2017); Lian et al. (2017).

Appropriately choosing the learning rate that scale coordinates of the gradient and the way of updating them are crucial issues driving the performance of first Duchi et al. (2011); Kingma & Ba (2014) and second order optimization procedures Peyghami & Tarzanagh (2015). Indeed, the understanding that adaptation of the learning rate is advantageous, particularly on a per parameter basis dynamically, led to the development of a family of widely-used adaptive gradient methods including ADAGRAD Duchi et al. (2011), RMSPROP Tieleman & Hinton, and ADAM Kingma & Ba (2014). The ADAM optimizer computes adaptive learning rates for different parameters from estimates of first and second moments of the gradients and performs a local optimization. Numerical results show that ADAM can achieve significantly better performance compared to ADAGRAD, RMSPROP and other gradient descent procedures when the gradients are sparse, or in general small in magnitude. However, its performance has been observed to deteriorate in settings where the loss functions are nonconvex and gradients are dense. Further, there is currently a gap in the theoretical understanding of these methods, especially in the nonconvex and stochastic setting Ward et al. (2018); Zeng & Yin (2018).

## 1.1 CONTENTS AND CONTRIBUTIONS

In this paper, we develop and analyze a new consensus-based distributed adaptive moment estimation (DADAM) method that incorporates decentralized optimization and uses a variant of the adaptive moment estimation methods Duchi et al. (2011); Kingma & Ba (2014); Tieleman & Hinton; McMahan & Streeter (2010). Existing distributed stochastic and adaptive gradient methods for deep learning are mostly designed for central network topology Dean et al. (2012); Li et al.. The main bottleneck of such a topology lies on the communication overload on the central node, since all nodes need to concurrently communicate with it. Hence, performance can be significantly degraded when network bandwidth is limited. These considerations motivate us to study an adaptive algorithm for network topologies, where all nodes can only communicate with their neighbors and none of the nodes is designated as "central". Therefore, the proposed method is suitable for large scale machine learning problems, since it enables both data parallelization and decentralized computation.

Next, we briefly summarize the main technical contributions of the work.

- Our first main result (Theorem 4) provides guarantees of DADAM for constrained convex minimization problems defined over a closed convex set $\mathscr{X}$. We provide the convergence bound in terms of *dynamic* regret and show that when the data features are sparse and have bounded gradients, our algorithm's regret bound can be considerably better than the ones provided by standard mirror descent and gradient descent methods Nedic & Ozdaglar (2009); Hall & Willett (2015); Besbes et al. (2015). It is worth mentioning that the regret bounds provided for adaptive gradient methods Duchi et al. (2011) are static and our results generalize them to dynamic settings.

- In Theorem 5, we give a new *local regret* analysis for distributed gradient-based algorithms for nonconvex minimization problems computed over a network of agents. Specifically, we prove that under certain regularity conditions, DADAM can achieve a local regret bound of order $\tilde{O}(\frac{1}{T})$ for nonconvex distributed optimization. To the best of our knowledge, rigorous extensions of existing adaptive gradient methods to the distributed nonconvex setting considered in this work do not seem to be available.

- In this paper, we also present regret analysis for distributed stochastic optimization problems computed over a network of agents. Theorems 8 and 10 provide regret bounds of DADAM for minimization problem equation 3 with stochastic gradients and indicate that the result of Theorems 4 and 5 holds true in expectation. Further, in Corollary 11 we show that DADAM can achieve a local regret bound of order $O(\frac{\xi^2}{\sqrt{nT}} + \frac{1}{T})$ for nonconvex distributed stochastic optimization where $\xi$ is an upper bound on the variance of the stochastic gradient. Hence, DADAM outperforms centralized adaptive algorithms such as ADAM for certain realistic classes of loss functions when $T$ is sufficiently large.

The remainder of the paper is organized as follows. Section 2 gives a detailed description of DADAM, while Section 3 establishes its theoretical results. Section 4 explains a network correction technique for our proposed algorithm. Section 5 illustrates the proposed framework on a number of synthetic and real data sets. Finally, Section 6 concludes the paper.

The detailed proofs of the main results established are delegated to the Appendix.

## 1.2 MATHEMATICAL PRELIMINARIES AND NOTATIONS.

Throughout the paper, $\mathbb{R}^p$ denotes the real coordinate space of $p$ dimensions. For any pair of vectors $x, y \in \mathbb{R}^p$, $\langle x, y \rangle$ indicates the standard Euclidean inner product. We denote the $\ell_1$ norm by $\|X\|_1 = \sum_{ij} |x_{ij}|$, the infinity norm by $\|X\|_\infty = \max_{ij} |x_{ij}|$, and the Euclidean norm by $\|X\| = \sqrt{\sum_{ij} |x_{ij}|^2}$. The above norms reduce to the vector norms if $X$ is a vector. The diameter of the set $\mathscr{X}$ is given by

$$\gamma_\infty = \sup_{x, y \in \mathscr{X}} \|x - y\|_\infty. \tag{1}$$

Let $\mathscr{S}_+^p$ be the set of all positive definite $p \times p$ matrices. $\Pi_{\mathscr{X}, A}[x]$ denotes the Euclidean projection of a vector $x$ onto $\mathscr{X}$ for $A \in \mathscr{S}_+^p$:

$$\Pi_{\mathscr{X}, A}[x] = \arg\min_{y \in \mathscr{X}} \|A^{\frac{1}{2}} (x - y)\|.$$

The subscript $t$ is often used to denote the time step while $y_{i,t,d}$ stands for the $d$-th element of $y_{i,t}$. Further, $y_{i,1:t,d} \in \mathbb{R}^t$ is given by

$$y_{i,1:t,d} = [y_{i,1,d}, y_{i,2,d}, \ldots, y_{i,t,d}]^\top.$$

We let $g_{i,t}$ to denote the gradient of $f$ at $x_{i,t}$. The $i$-th largest singular value of matrix $X$ is denoted by $\sigma_i(X)$. We denote the element in the $i$-th row and $j$-th column of matrix $X$ by $[X]_{ij}$. In several theorems, we consider a connected undirected graph $\mathscr{G} = (\mathscr{V}, \mathscr{E})$ with nodes $\mathscr{V} = \{1, 2, \ldots, n\}$ and edges $\mathscr{E}$. The matrix $W \in \mathbb{R}^{n \times n}$ is often used to denote a symmetric mixing matrix Shi et al. (2015) of graph $\mathscr{G}$. Indeed, for any agent $i$, we assign a positive weight $[W]_{ij}$ for the information received from agent $j \neq i$ so that $[W]_{ij} > 0$ if and only if $(i, j) \in \mathscr{E}$ and

$$\sum_{i=1}^{n} [W]_{ij} = \sum_{j=1}^{n} [W]_{ij} = 1. \tag{2}$$

The role of $W$ is the similar as that in decentralized gradient descent methods Yuan et al. (2016) and average consensus Xiao et al. (2007). It has a few common choices, which can significantly affect performance (see, Shi et al. (2015)). Throughout this paper, we consider Metropolis constant edge weight matrix Xiao & Boyd (2004); Xiao et al. (2007) defined as follows

$$[\hat{W}]_{ij} = \begin{cases} \frac{1}{\max\{\deg(i), \deg(j)\} + \varepsilon}, & \text{if } (i, j) \in \mathscr{E}, \\ 0, & \text{if } (i, j) \notin \mathscr{E} \text{ and } i \neq j, \\ 1 - \sum_{k \in \mathscr{V}} [\hat{W}]_{ik}, & \text{if } i = j, \end{cases}$$

for some small positive $\varepsilon > 0$. When $\hat{W}$ is chosen according this scheme, $W = \frac{I + \hat{W}}{2}$ is found to be very efficient. Note, the above doubly stochastic matrix implies uniqueness of $\sigma_1(W) = 1$ and warrants that other singular value of $W$ are strictly less than one in magnitude.

The Hadamard (entrywise) and Kronecker product are denoted by $\odot$ and $\otimes$, respectively. Finally, the expectation operator is denoted by $\mathbb{E}$.

## 2 PROBLEM FORMULATION AND ALGORITHM

In this section, we propose a new optimization method for online optimization that employs data parallelization and decentralized computation over a network of agents. In our new structure, given a connected undirected graph $\mathscr{G} = (\mathscr{V}, \mathscr{E})$, we let each node $i \in \mathscr{V}$ at time $t \in \{1, \ldots, T\}$ hold its own measurement and training data $m_i$, and set $f_{i,t}(x) = \frac{1}{m_i} \sum_{j=1}^{m_i} f_{i,t}^j(x)$. We also let each agent $i$ hold a local copy of the global variable $x$ at time $t \in \{1, \ldots, T\}$, which is denoted by $x_{i,t} \in \mathbb{R}^p$. With this setup, we present a distributed adaptive gradient method for solving the minimization problem

$$\underset{x \in \mathscr{X}}{\text{minimize}} \quad F(x) = \frac{1}{n} \sum_{t=1}^{T} \sum_{i=1}^{n} f_{i,t}(x), \tag{3}$$

where $f_{i,t} : \mathscr{X} \rightarrow \mathbb{R}$ is a continuously differentiable mapping on the convex set $\mathscr{X}$.

It is known that it is impossible to achieve a sub-linear dynamic regret bound Zhang et al. (2017), because of the arbitrary fluctuations in problem equation 3. We want to characterize the hardness of the problem via a complexly measure that captures the pattern of the minimizer sequence $\{x_t^*\}_{t=1}^T$, where $x_t^* = \arg\min_{x \in \mathscr{X}} f_t(x)$. Subsequently, we would like to provide a regret bound in terms of

$$D_{T,d} = \sum_{t=1}^{T-1} |x_{t+1,d}^* - x_{t,d}^*| \qquad \text{for} \qquad d \in \{1, ..., p\}, \tag{4}$$

which represents the variations in $\{x_t^*\}_{t=1}^T$.

DADAM uses a new distributed adaptive gradient method in which a group of $n$ agents seeking to solve a sequential version of problem equation 3. Here, we assume that each component function $f_{i,t} : \mathscr{X} \rightarrow \mathbb{R}$ becomes *only* available to agent $i \in \mathscr{V}$, after having made its decision at time $t \in \{1, ..., T\}$. In the $t$-th, the $i$-th agent chooses a point $x_{i,t}$ corresponding to what it considers the network as a whole should have selected. After committing to this choice, the agent has access to a cost function $f_{i,t} : \mathscr{X} \rightarrow \mathbb{R}$ and the network cost is then given by $f_t(x) = \frac{1}{n}\sum_{i=1}^n f_{i,t}(x)$. Note that this function is not known to any of the agents and is not available at any single location.

The procedure of our proposed method is outlined in Algorithm 1.

---

**Algorithm 1:** A distributed adaptive moment estimation method (DADAM)

---

**input :** $x_1 \in \mathscr{X}$, a positive sequence $\{\alpha_t\}_{t=1}^T$, exponential decay rates for the moment estimate $\beta_1, \beta_2, \beta_3 \in [0,1)$, and a mixing matrix $W$ satisfying equation 2;

1   for all $i \in \mathscr{V}$, initialize moment vectors $m_{i,0} = \upsilon_{i,0} = \hat{\upsilon}_{i,0} = 0$ and $x_{i,1} = x_1$;
2   **for** $i \in \mathscr{V}$ **do**
3     **for** $t \leftarrow 1$ **to** $T$ **do**
4       $g_{i,t} = \nabla f_{i,t}(x_{i,t})$;
5       $m_{i,t} = \beta_1 m_{i,t-1} + (1-\beta_1)g_{i,t}$;
6       $\upsilon_{i,t} = \beta_2 \upsilon_{i,t-1} + (1-\beta_2)g_{i,t} \odot g_{i,t}$;
7       $\hat{\upsilon}_{i,t} = \beta_3 \upsilon_{i,t} + (1-\beta_3)\max(\hat{\upsilon}_{i,t-1}, \upsilon_{i,t})$;
8       $x_{i,t+\frac{1}{2}} = \sum_{j=1}^n [W]_{ij}x_{j,t}$;
9       $x_{i,t+1} = \Pi_{\mathscr{X}, \sqrt{\mathrm{diag}(\hat{\upsilon}_{i,t})}} \left[ x_{i,t+\frac{1}{2}} - \alpha_t \frac{m_{i,t}}{\sqrt{\hat{\upsilon}_{i,t}}} \right]$;

**output:** resulting parameter $\bar{x} = \frac{1}{n}\sum_{i=1}^n x_{i,T+1}$

---

It is worth mentioning that DADAM computes adaptive learning rates from estimates of both first and second moments of the gradients similar to Duchi et al. (2011); Tieleman & Hinton; Kingma & Ba (2014); Reddi et al. (2018). However, DADAM uses a larger learning rate in comparison to AMSGrad Reddi et al. (2018) and yet incorporates the intuition of slowly decaying the effect of previous gradients on the learning rate. The key difference of DADAM with ADAM and AMSGrad is that it maintains the maximum of all second moment estimates of the gradient vectors until time step $t$ and uses

$$\hat{\upsilon}_{i,t} = \beta_3 \upsilon_{i,t} + (1-\beta_3)\max(\hat{\upsilon}_{i,t-1}, \upsilon_{i,t}),$$

for normalizing the running average of the gradient instead of $\upsilon_{i,t}$ in ADAM and $\max(\hat{\upsilon}_{i,t-1}, \upsilon_{i,t})$ in AMSGrad. The learning rate $\hat{\upsilon}_{i,t}$ is an important component of the DADAM framework, since it enables us to develop a convergent adaptive method similar to AMSGrad while maintaining the efficiency of ADAM.

Next, we provide the notion of regret which measures the performance of DADAM against a sequence of local minimizers.

In the framework of online convex optimization, the performance of algorithms is assessed by regret that measures how competitive the algorithm is with respect to the best fixed solution Mateos-Núñez & Cortés (2014); Hazan et al. (2016). However, the notion of regret fails to illustrate the performance of online algorithms in a dynamic setting. To overcome this issue, we consider a more stringent

metric–dynamic regret Hall & Willett (2015); Besbes et al. (2015); Zinkevich (2003), in which the cumulative loss of the learner is compared against the minimizer sequence $\{x_t^*\}_{t=1}^T$, i.e.,

$$\mathbf{Reg}_T^C := \frac{1}{n}\sum_{i=1}^n\sum_{t=1}^T f_{i,t}(x_{i,t}) - \sum_{t=1}^T f_t(x_t^*),$$

where $x_t^* = \arg\min_{x \in \mathcal{X}} f_t(x)$.

On the other hand, in the framework of nonconvex optimization, it is usual to state convergence guarantees of an algorithm towards an $\varepsilon$-approximate stationary point–that is, there exist some iterates $x_{i,t}$ for which $\|\nabla f_t(x_{i,t})\| \leq \varepsilon$. Inspired by Hazan et al. (2017), we next provide the definition of projected gradient and introduce *local regret*, a new notion of regret which quantifies the moving average of gradients over network.

**Definition 1.** (Local Regret). *Assume $f_i : \mathcal{X} \to \mathbb{R}$ is a differentiable function on a closed convex set $\mathcal{X} \subseteq \mathbb{R}^p$. Given a step-size $\alpha > 0$, we define $G_{\mathcal{X}}(x, f_i, \alpha) : \mathcal{X} \to \mathbb{R}^p$ the projected gradient of $f_i$ at x, by*

$$G_{\mathcal{X}}(x, f_i, \alpha) = \frac{\sqrt{\hat{v}_i}}{\alpha}(x - x_i^+), \qquad \forall i \in \mathcal{V}, \tag{5}$$

*where*

$$x_i^+ = \arg\min_{y \in \mathcal{X}}\{\langle y, \frac{m_i}{\sqrt{\hat{v}_i}}\rangle + \frac{1}{2\alpha}\|y - \sum_{j=1}^n [W]_{ij}x_j\|^2\}. \tag{6}$$

*Then, the local regret of an online algorithm is given by*

$$\mathbf{Reg}_T^N := \frac{1}{n}\sum_{i=1}^n \min_{t \in \{1,\dots,T\}} \|G_{\mathcal{X}}(x_{i,t}, \bar{f}_{i,t}, \alpha_t)\|^2,$$

*where $\bar{f}_{i,t}(x_{i,t}) = \frac{1}{t}\sum_{s=1}^t f_{i,s}(x_{i,t})$ is an aggregate loss.*

We analyze the convergence of DADAM as applied to minimization problem equation 3 using regrets $\mathbf{Reg}_T^C$ and $\mathbf{Reg}_T^N$. It is worth to mention that DADAM is initialized at $x_{i,t} = 0$ to keep the presentation of the convergence analysis clear. In general, any initialization can be selected for implementation purposes.

## 3 CONVERGENCE ANALYSIS

In this section, our aim is to establish convergence properties of DADAM under the following assumptions:

**Assumption 2.** *For all $i \in \mathcal{V}$ and $t \in \{1,\dots,T\}$, the function $f_{i,t}(x)$ is continuously differentiable over $\mathcal{X}$, and has Lipschitz continuous gradient on this set, i.e. there exists a constant $\rho \geq 0$ so that*

$$\|\nabla f_{i,t}(x) - \nabla f_{i,t}(y)\| \leq \rho\|x - y\|, \qquad \forall x, y \in \mathcal{X}.$$

*Further, $f_{i,t}(\cdot)$ is Lipschitz continuous on $\mathcal{X}$ with a uniform constant $L \geq 0$, i.e.,*

$$|f_{i,t}(x) - f_{i,t}(y)| \leq L\|x - y\|, \qquad \forall x, y \in \mathcal{X}. \tag{7}$$

There are many cost functions $f_{i,t}$ that satisfy this type of Lipschitz condition. For example, it holds for any convex function on a compact set $\mathcal{X}$, or for any polyhedral function on an arbitrary domain Duchi et al. (2011).

**Assumption 3.** *For all $i \in \mathcal{V}$ and $t \in \{1,\dots,T\}$, the stochastic gradient, as notation we use bold letters $\boldsymbol{g} = \nabla f_{i,t}(x_{i,t})$, satisfies*

$$\mathbb{E}\left[\nabla f_{i,t}(x_{i,t})\big|\mathscr{F}_{t-1}\right] = \nabla f_{i,t}(x_{i,t}), \qquad \mathbb{E}\left[\|\nabla f_{i,t}(x_{i,t})\|^2\big|\mathscr{F}_{t-1}\right] \leq \xi^2,$$

*where $\mathscr{F}_t$ is the $\xi$-field containing all information prior to the outset of round $t + 1$.*

## 3.1 Convex Case

Next, we focus on the case where for all $i \in \mathcal{V}$ and $t \in \{1, \ldots, T\}$, the agent $i$ at time $t$ has access to the exact gradient $g_{i,t} = \nabla f_{i,t}(x_{i,t})$. The following results apply to convex problems, as well as their stochastic variants in a dynamic environment.

**Theorem 4.** *Suppose that the parameters $\beta_1, \beta_2 \in [0,1)$ satisfy $\eta = \frac{\beta_1}{\sqrt{\beta_2}} < 1$. Let $\beta_{1,t} = \beta_1 \lambda^{t-1}, \lambda \in (0,1)$ and $\|\nabla f_{i,t}(x)\|_\infty \leq G_\infty$ for all $t \in \{1, \ldots, T\}$. Then, using a step-size $\alpha_t = \frac{\alpha}{\sqrt{t}}$ for the sequence $x_{i,t}$ generated by Algorithm 1, we have*

$$\mathbf{Reg}_T^C \leq \frac{(1-\beta_1)\alpha\sqrt{1+\log T}}{2\sqrt{n}\sqrt{(1-\beta_2)}} \sum_{d=1}^{p} \|g_{1:T,d}\| + \sum_{d=1}^{p} \frac{G_\infty \gamma_\infty}{(1-\beta_1)(1-\lambda)}$$
$$+ \sum_{d=1}^{p} \frac{\gamma_\infty(\gamma_\infty + D_{T,d})}{\sqrt{n}(1-\beta_1)\alpha} \sqrt{T\hat{v}_{T,d}} + \frac{4\alpha\sqrt{1+\log T}\sum_{d=1}^{p}\|g_{1:T,d}\|}{(1-\sigma_2(W))\sqrt{(1-\beta_1)}\sqrt{(1-\eta)}\sqrt{(1-\beta_2)}}.$$

Next, we analyze the stochastic convex setting and extend the result of Theorem 4 to the noisy case where agents have access to stochastic gradients of the objective function equation 3.

**Theorem 5.** *Suppose that Assumption 3 holds. Further, the parameters $\beta_1, \beta_2 \in [0,1)$ satisfy $\eta = \frac{\beta_1}{\sqrt{\beta_2}} < 1$. Let $\beta_{1,t} = \beta_1 \lambda^{t-1}, \lambda \in (0,1)$. Then, using a step-size $\alpha_t = \frac{\alpha}{\sqrt{t}}$ for the sequence $x_{i,t}$ generated by Algorithm 1, we have*

$$\mathbb{E}\left[\mathbf{Reg}_T^C\right] \leq \frac{(1-\beta_1)\alpha\sqrt{1+\log T}}{2\sqrt{n}\sqrt{(1-\beta_2)}} \sum_{d=1}^{p} \mathbb{E}\left[\|g_{1:T,d}\|\right] + \sum_{d=1}^{p} \frac{\xi\gamma_\infty}{(1-\beta_1)(1-\lambda)}$$
$$+ \sum_{d=1}^{p} \frac{\gamma_\infty(\gamma_\infty + D_{T,d})}{\sqrt{n}(1-\beta_1)\alpha} \sqrt{T}\mathbb{E}\left[\sqrt{\hat{v}_{T,d}}\right] + \frac{4\alpha\sqrt{1+\log T}\sum_{d=1}^{p}\mathbb{E}\left[\|g_{1:T,d}\|\right]}{(1-\sigma_2(W))\sqrt{(1-\beta_1)}\sqrt{(1-\eta)}\sqrt{(1-\beta_2)}}.$$

**Remark 6.** *Theorems 4 and 5 show that the regret bound of DADAM can be considerably better than the ones provided by standard mirror descent and gradient descent methods for both centralized and decentralized settings Zinkevich (2003); Hall & Willett (2015); Besbes et al. (2015); Shahrampour & Jadbabaie (2018) because*

$$\sum_{d=1}^{p} \|g_{1:T,d}\| = \sum_{d=1}^{p} \sqrt{g_{1,d}^2 + g_{2,d}^2 + \cdots + g_{T,d}^2} \leq \sum_{d=1}^{p} \sqrt{G_\infty^2 + G_\infty^2 + \cdots + G_\infty^2}$$
$$= \sum_{d=1}^{p} \sqrt{T}G_\infty = pG_\infty\sqrt{T},$$

$$\sum_{d=1}^{p} \sqrt{T\hat{v}_{T,d}} \leq \sum_{d=1}^{p} \sqrt{T}G_\infty \leq pG_\infty\sqrt{T}.$$

It is easy to show that the regret of DADAM is upper bounded by $O(\frac{G_\infty D_T \sqrt{T}}{1-\sigma_2(W)})$ where $D_T = \max_{d \in \{1,\ldots,p\}} D_{T,d}$. Indeed, the term $\sum_{t=1}^{T} |g_{i,t,d}|/\sqrt{t}$ in the proof of Lemma 14 can be bounded by $O(G_\infty\sqrt{T})$ instead of $O(G_\infty\sqrt{T\log T})$. Hence, the regret of DADAM is upper bounded by minimum of $O(\frac{G_\infty D_T \sqrt{T}}{1-\sigma_2(W)})$ and the bound presented in Theorems 4 and 5, and thus the worst case dependence on $T$ is $\sqrt{T}$ rather than $\sqrt{T\log T}$.

**Remark 7.** *We note that in the static setting, i.e. $D_T = 0$, the regret of DADAM is upper bounded by*

$$O(\frac{G_\infty\sqrt{T}}{1-\sigma_2(W)}),$$

*where $1 - \sigma_2(W)$ is the spectral gap of the network.*

## 3.2 NONCONVEX CASE

In this section, we provide convergence guarantees for DADAM for the nonconvex minimization problem equation 3 defined over a closed convex set $\mathcal{X}$. To do so, we use the projection map $\Pi_{\mathcal{X}}$ instead of $\Pi_{\mathcal{X}, \sqrt{\text{diag}(\hat{v}_{i,t})}}$ for updating parameters $x_{i,t}$ for $t \in \{1, \ldots, T\}$, $i \in \mathcal{V}$ (see, Algorithm 1 for details).

Next, we derive an upper and lower bound for the second moment of gradient, $v_{i,t}$. Assume that there exists a positive number $\bar{v}$ such that

$$\max_{d \in \{1, \ldots, p\}} g_{i,t,d} \leq \bar{v}. \tag{8}$$

Now, using the update rule of $v_{i,t}$, we have $v_{i,t} = (1 - \beta_2) \sum_{l=1}^{t} \beta_2^{t-l} g_{i,l}^2$. This together with equation 8 imply that

$$\sqrt{v_{i,t}} \leq \max_{d \in \{1, \ldots, p\}} \sqrt{v_{i,t,d}} = \max_{d \in \{1, \ldots, p\}} \sqrt{(1 - \beta_2) \sum_{l=1}^{t} \beta_2^{t-l} g_{i,l,d}^2}$$

$$\leq \bar{v} \sqrt{(1 - \beta_2) \sum_{l=1}^{t} \beta_2^{t-l}} \leq \bar{v}. \tag{9}$$

On the other hand, since $v_{i,t,d}$ is non-decreasing and $g_{i,t,d}$ is required to be element-wise non-zero, we have

$$v_{i,t,d} \geq \underline{v}^2 > 0, \tag{10}$$

for some constant $\underline{v}$.

Now, using the update rule of $\hat{v}_{i,t}$ along with equation 10 and equation 9, we have

$$\underline{v}^2 \leq \beta_3 \underline{v}^2 + (1 - \beta_3) \underline{v}^2$$
$$\leq \hat{v}_{i,t} = \beta_3 v_{i,t} + (1 - \beta_3) \max(\hat{v}_{i,t-1}, v_{i,t})$$
$$\leq \beta_3 \bar{v}^2 + (1 - \beta_3) \bar{v}^2 \leq \bar{v}^2. \tag{11}$$

**Theorem 8.** *Suppose that Assumption 2 holds. Further, the parameters $\beta_1, \beta_2 \in [0, 1)$ satisfy $\eta = \frac{\beta_1}{\sqrt{\beta_2}} < 1$. Choose the positive sequence $\{\alpha_t\}_{t=1}^{T}$ such that $0 < \alpha_t \leq \frac{(2-\beta_1)\underline{v}^2}{\rho \bar{v}}$ with $\alpha_t < \frac{(2-\beta_1)\underline{v}^2}{\rho \bar{v}}$ for at least one t. Then, for the sequence $x_{i,t}$ generated by Algorithm 1, we have*

$$\mathbf{Reg}_T^N \leq \frac{1}{\vartheta_t} \left[ (2 + \log T) 2L \max_{t \in \{2, \ldots, T\}} \frac{2\sqrt{n}}{(1-\eta)\sqrt{(1-\beta_2)}} \sum_{s=0}^{t-1} \alpha_s \sigma_2^{t-s-1}(W) \right.$$

$$\left. + \sum_{t=1}^{T} \frac{\alpha_t \beta_{1,t} \bar{v}}{2(1-\beta_{1,t})(1-\eta)^2(1-\beta_2)} \right], \tag{12}$$

*where $\vartheta_t = \sum_{t=1}^{T} \left[ \frac{(2-\beta_1)\alpha_t}{2\bar{v}} - \frac{\rho \alpha_t^2}{2\underline{v}^2} \right]$.*

The following corollary shows that DADAM using a certain step-size leads to a near optimal regret bound for nonconvex functions.

**Corollary 9.** *Under the same conditions of Theorem 8, using the step-sizes $\alpha_t = \frac{(2-\beta_1)\underline{v}^2}{2\rho \bar{v}}$ and $\beta_{1,t} = \beta_1 \lambda^{t-1}, \lambda \in (0, 1)$ for all $t \in \{1, \ldots, T\}$, we have*

$$\mathbf{Reg}_T^N \leq \left( \frac{2\bar{v}^2}{(2-\beta_1)(1-\beta_1)(1-\eta)^2(1-\beta_2)(1-\lambda)} \right) \frac{1}{T}$$

$$+ \left( \frac{16\sqrt{n}\bar{v}L}{(2-\beta_1)(1-\eta)\sqrt{(1-\beta_2)}(1-\sigma_2(W))} \right) \frac{(2 + \log T)}{T}. \tag{13}$$

To complete the analysis of our algorithm in the nonconvex setting, we provide the regret bound for DADAM, when stochastic gradients are accessible to the learner.

**Theorem 10.** *Suppose that Assumption 2 holds. Further, the parameters $\beta_1, \beta_2 \in [0,1)$ satisfy $\eta = \frac{\beta_1}{\sqrt{\beta_2}} < 1$. Choose the positive sequence $\{\alpha_t\}_{t=1}^T$ such that $0 < \alpha_t \leq \frac{(2-\beta_1)\underline{v}^2}{\rho\bar{v}}$ with $\alpha_t < \frac{(2-\beta_1)\underline{v}^2}{\rho\bar{v}}$ for at least one t. Then, for the sequence $x_{i,t}$ generated by Algorithm 1, we have*

$$\mathbb{E}\left[\mathbf{Reg}_T^N\right] \leq \frac{1}{\vartheta_t}\left[(2+\log T)2L \max_{t\in\{2,\dots,T\}} \frac{2\sqrt{n}}{(1-\eta)\sqrt{(1-\beta_2)}} \sum_{s=0}^{t-1} \alpha_s \sigma_2^{t-s-1}(W)\right.$$

$$\left.+ \sum_{t=1}^T \frac{\alpha_t \beta_{1,t}\bar{v}}{2(1-\beta_{1,t})(1-\eta)^2(1-\beta_2)} + \frac{\bar{v}\xi^2}{\underline{v}^2(1-\beta_1)}\sum_{t=1}^T \alpha_t\right], \tag{14}$$

*where $\vartheta_t = \sum_{t=1}^T \left[\frac{(2-\beta_1)\alpha_t}{2\bar{v}} - \frac{\rho\alpha_t^2}{2\underline{v}^2}\right].$*

### 3.2.1 WHEN DOES DADAM OUTPERFORM ADAM?

We next theoretically justify the potential advantage of the proposed decentralized algorithm DADAM over centralized adaptive moment estimation methods such as ADAM . More specifically, the following corollary shows that when $T$ is sufficiently large, the $\frac{1}{T}$ term will be dominated by the $\frac{1}{\sqrt{nT}}$ term which leads to a $\frac{1}{\sqrt{nT}}$ convergence rate.

**Corollary 11.** *Suppose that Assumptions 2 and 3 hold. Moreover, the parameters $\beta_1, \beta_2 \in [0,1)$ satisfy $\eta = \frac{\beta_1}{\sqrt{\beta_2}} < 1$ and $\beta_{1,t} = \beta_1\lambda^{t-1}, \lambda \in (0,1)$. Choose the step-size sequence as $\alpha_t = \frac{\alpha}{\sqrt{nT}}$ with $\alpha = \frac{(2-\beta_1)\underline{v}^2}{\rho\bar{v}}$. Then, for the sequence $x_{i,t}$ generated by Algorithm 1, we have*

$$\frac{\mathbb{E}\left[\mathbf{Reg}_T^N\right]}{T} \leq \left(\frac{8\bar{v}\alpha}{\underline{v}^2(1-\beta_1)}\right)\frac{\xi^2}{\sqrt{nT}} + 2\left(f_1(x_1) - f_1(x_1^*)\right)\frac{1}{T}, \tag{15}$$

*if the total number of time steps $T$ satisfies*

$$T \geq (I_1 + I_2), \tag{16a}$$

$$T \geq \max\left\{\frac{4\rho^2\bar{v}^2}{n\underline{v}^4(2-\beta_1)^2}, \frac{4\bar{v}^2 n}{(2-\beta_1)^2}\right\}, \tag{16b}$$

*where*

$$I_1 = \frac{\underline{v}^2}{2(1-\eta)^2(1-\beta_2)(1-\lambda)\xi^2}, \qquad I_2 = \frac{16nL^2\underline{v}^4(1-\beta_1)^2}{(1-\eta)^2(1-\beta_2)(1-\sigma_2(W))^2\bar{v}^2\xi^4}.$$

Let $\varepsilon$-approximation solution of equation 3 be defined by $\frac{\mathbb{E}\left[\mathbf{Reg}_T^N\right]}{T} \leq \varepsilon$. Corollary 11 indicates that the total computational complexity of DADAM to achieve an $\varepsilon$-approximation solution is bounded by $O(\frac{1}{\varepsilon^2})$. The key setup which leads to this regret bound is that we do not use the boundedness assumption for domain or gradient. These assumptions can simplify the proof but lose some of the sophisticated structures in the problem.

## 4 AN EXTENSION OF DADAM WITH A CORRECTED UPDATE RULE

Compared to classical centralized algorithms, decentralized algorithms encounter more restrictive assumptions and typically worse convergence rates. In order to improve the convergence rate, Shi et al. (2015) introduced a corrected decentralized gradient method called EXTRA, which has a linear rate of convergence if the objective function is strongly convex. Similar to EXTRA, we provide next a corrected update rule for $i \in \mathcal{V}$, given by

$$x_{i,t+1}^{\text{C-DADAM}} = x_{i,t+1} + \underbrace{\sum_{s=0}^{t-1}\sum_{j=1}^n [W - \hat{W}]_{ij}x_{j,s}}_{\text{correction}}, \quad t = 0, 1, \dots. \tag{17}$$

where $x_{i,t+1}$ is generated by Algorithm 1.

Our numerical results on some test problems show the efficiency and effectiveness of the corrected DADAM (C-DADAM) in practice. Note that a C-DADAM update is a DADAM update with a cumulative correction term. The summation in equation 17 is necessary, since each individual term $\sum_{j=1}^{n}[W - \hat{W}]_{ij}x_{j,s}$ is asymptotically vanishing and the terms must work cumulatively.

## 5 NUMERICAL RESULTS

In this section, we present numerical results for Algorithm 1 (DADAM), on both synthetic and real data sets. The employed data sets and the code will be made available on github upon acceptance of this manuscript. We first present numerical results on a sparse binary classification problem, comparing the performance of DADAM to that of standard decentralized gradient descent (DGD) Nedic & Ozdaglar (2009); Yuan et al. (2016) and its corrected version EXTRA Shi et al. (2015). In Section 5.2, we demonstrate the efficacy of DADAM in comparison with a decentralized ADAM (Algorithm 1 with $\beta_3 = 1$) and the recently proposed distributed federated averaging SGD (FedAvg) algorithm on benchmark datasets such as MNIST and CIFAR-10. All algorithms have been implemented in a Mac machine equipped with a 1.8 GHz Intel Core i5 processor and 8 GB 1600 MHz DDR3 of memory.

### 5.1 BINARY CLASSIFICATION BASED ON SYNTHETIC DATA IN THE STATIC ENVIRONMENT

Consider the following online distributed learning setting: at each time $t$, $m_i$ randomly generated data points are given to every agent $i$ in the form of $(y_{t,i,j}, \boldsymbol{x}_{t,i,j})$, where $y_{t,i,j} \in \{\pm 1\}$ is a binary label and $\boldsymbol{x}_{t,i,j} \in \mathbb{R}^p$ is a feature vector for $j = 1, \ldots, m_i$. Our goal is to train a classifier $\theta \in \mathbb{R}^p$ such that for an arbitrary new feature vector $\hat{\boldsymbol{x}}$ it assigns $\hat{y} = \text{sign}(\langle \theta, \hat{\boldsymbol{x}} \rangle)$. We design decentralized convex logistic and nonconvex sigmoid loss functions with $\ell_2$ regularization as follows:

$$f_{i,t}^1(\boldsymbol{\theta}) = \frac{1}{m_i} \sum_{j=1}^{m_i} \left(1 + \exp(10\langle \boldsymbol{x}_{t,i,j}, \boldsymbol{\theta}\rangle y_{t,i,j})\right)^{-1} + \nu\|\boldsymbol{\theta}\|_2^2,$$

$$f_{i,t}^2(\boldsymbol{\theta}) = \frac{1}{m_i} \sum_{j=1}^{m_i} \log(1 + \exp(-\langle \boldsymbol{x}_{t,i,j}, \boldsymbol{\theta}\rangle y_{t,i,j})) + \nu\|\boldsymbol{\theta}\|_2^2, \tag{18}$$

where $\nu = 1/m_i$ and $m_i$ is the size of the training data on node $i$.

For $\mathscr{X}$, we consider the $\ell_1$ ball $\mathscr{X}_{\ell_1} = \{\boldsymbol{\theta} \in \mathbb{R}^p : \|\boldsymbol{\theta}\|_1 \leq r\}$, when a sparse classifier is preferred.

We show results for the decentralized problem solved by DADAM, C-DADAM, DGD and EXTRA over a medium-scale network. We have implemented EXTRA and DGD with their default settings. The connected network is randomly generated with $n = 50$ agents and connectivity ratio $r = 0.2$. Each agent holds 10 samples, i.e., $m_i = 10, \forall i \in \mathscr{V}$. The agents shall collaboratively obtain $p$ coefficients via loss functions equation 18. All samples are randomly generated, and the reference classifier $\theta^*$ is pre-computed using a centralized method. As it is easy to implement in practice, we use the Metropolis constant edge weight matrix $W$ Shi et al. (2015) in our experimental evaluation [1]. We have tested the DADAM and C-DADAM with $\beta_1 = \beta_2 = \beta_3 = 0.9$. The numerical results are illustrated in Figure 1. It can be seen that DADAM and C-DADAM outperform DGD and EXTRA, showing almost linear convergence in the convex setting to the reference logistic classifier $x^*$.

### 5.2 CIFAR AND MNIST EXPERIMENTS

Next, we present the experimental results using the CIFAR-10 image recognition dataset and the MNIST digit recognition task. The model architecture for training CNN to classify CIFAR10 dataset was taken from the TensorFlow tutorial [2]. Also, the model for training a simple multilayer percepton (MLP) on the MNIST dataset was taken from `https://github.com/keras-team/keras/blob/master`.

---

[1] `http://www.math.ucla.edu/ wotaoyin/software.html`
[2] `http://www.tensorflow.org/tutorials`

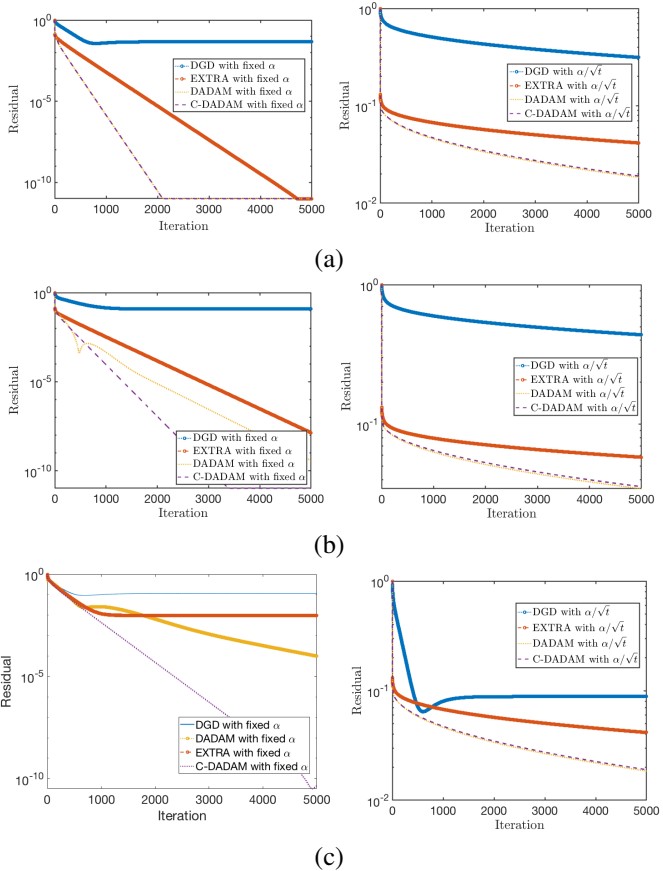

Figure 1: The residual $\frac{\|\theta_t - \theta^*\|}{\|\theta_0 - \theta^*\|}$ on iteration (time step $t \in \{1, \ldots, T\}$) for binary classification problem using convex and nonconvex (with $\alpha/\sqrt{t}$) loss functions. Constant $\alpha = \sigma_n/\rho$ is the theoretical critical step-size given for DGD Yuan et al. (2016). (a) classification with $p = 10$ and $\nu = 0$. (b) classification with $p = 20$ and $\nu = 0$. (c) classification with $p = 20$ and $\nu = 1/m_i$.

We compare the accuracy of DADAM with that of the decentralized ADAM (Algorithm 1 with $\beta_3 = 1$) and the Federated Averaging (FedAvg) algorithm McMahan et al. (2016) which also performs data parallelization without decentralized computation. The connected network is generated with $n = 5$ agents and connectivity ratio $r = 0.8$. The parameters for DADAM and ADAM are selected in a way similar to the previous experiments. In our implementation, we use same number of agents and choose $E = C = 1$ as the parameters in the FedAvg algorithm since it is close to a connected topology scenario as considered in the DADAM and ADAM. It can be easily seen from Figure 2 that DADAM and the decentralized ADAM ($\beta_3 = 1$) outperform FedAvg. Further, DADAM can achieve high accuracy in comparison with the decentralized ADAM and FedAvg.

## 6    CONCLUSION

A decentralized adaptive moment estimation method DADAM was proposed for the distributed learning of deep networks based on adaptive moment of first and second moment of estimations. Convergence properties of the proposed algorithm were established for convex (and dynamic) and nonconvex functions in both stochastic and deterministic settings. Numerical results on some datasets show the efficiency and effectiveness of the new proposed method in practice.

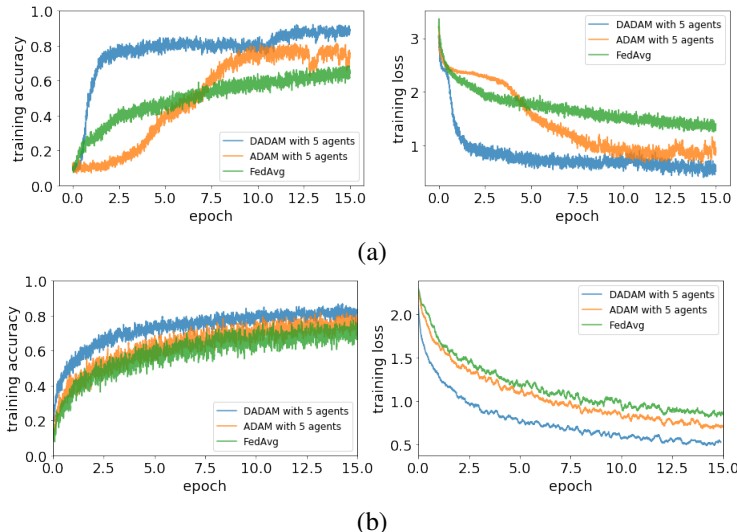

Figure 2: Training loss and accuracy over 15 epochs. (a) MNIST digit recognition task (b) CIFAR10 image recognition dataset.

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

Next, we establish a series of lemmas used in the proof of main theorems.

## 6.1 PROPERTIES OF DADAM

**Lemma 12.** *Beck & Teboulle (2003) Let $\mathscr{X}$ be a nonempty closed convex set in $\mathbb{R}^p$. Then, for any $d \in \mathscr{X}$, we have*

$$\langle x^* - d, a \rangle \leq \frac{1}{2}\|d - c\|^2 - \frac{1}{2}\|d - x^*\|^2 - \frac{1}{2}\|x^* - c\|^2,$$

*where*

$$x^* = \arg\min_{x \in \mathscr{X}}\{\langle a, x \rangle + \frac{1}{2}\|x - c\|^2\}.$$

**Lemma 13.** *McMahan & Streeter (2010) For any $A \in \mathscr{S}_+^p$ and convex feasible set $C \subset \mathbb{R}^p$, suppose $a_1 = \Pi_{C,A}[b_1], a_2 = \Pi_{C,A}[b_2]$, we have*

$$\|A^{\frac{1}{2}}(a_1 - a_2)\| \leq \|A^{\frac{1}{2}}(b_1 - b_2)\|.$$

**Lemma 14.** *For all $i \in \mathscr{V}$ if $\beta_1, \beta_2 \in [0,1)$ satisfy $\eta = \frac{\beta_1}{\sqrt{\beta_2}} < 1$, then we have*

$$\sum_{t=1}^{T}\sum_{d=1}^{p}\frac{\alpha_t m_{i,t,d}^2}{\hat{v}_{i,t,d}} \leq \frac{\alpha\sqrt{1+\log T}}{(1-\beta_1)(1-\eta)\sqrt{(1-\beta_2)}}\sum_{d=1}^{p}\|g_{i,1:T,d}\|,$$

*where $\alpha_t = \frac{\alpha}{\sqrt{t}}$ for all $t \in \{1,\ldots,T\}$.*

*Proof.* Using the update rules in Algorithm 1, we have

$$\sum_{t=1}^{T}\frac{\alpha m_{i,t,d}^2}{\sqrt{t}\hat{v}_{i,t,d}} = \sum_{t=1}^{T-1}\frac{\alpha m_{i,t,d}^2}{\sqrt{t}\hat{v}_{i,t,d}} + \frac{\alpha_T m_{i,T,d}^2}{\sqrt{(1-\beta_3)\max\{\hat{v}_{i,T-1,d}, v_{i,T,d}\} + \beta_3 v_{i,T,d}}}$$

$$\leq \sum_{t=1}^{T-1}\frac{\alpha m_{i,t,d}^2}{\sqrt{t}\hat{v}_{i,t,d}} + \frac{\alpha_T m_{i,T,d}^2}{\sqrt{(1-\beta_3)v_{i,T,d} + \beta_3 v_{i,T,d}}}$$

$$\stackrel{(i)}{=} \sum_{t=1}^{T-1}\frac{\alpha m_{i,t,d}^2}{\sqrt{t}\hat{v}_{i,t,d}} + \frac{\alpha(\sum_{l=1}^{T}(1-\beta_1)\beta_1^{T-l}g_{i,l,d})^2}{\sqrt{T\sum_{l=1}^{T}(1-\beta_2)\beta_2^{T-l}g_{i,l,d}^2}}$$

$$\stackrel{(ii)}{\leq} \sum_{t=1}^{T-1}\frac{\alpha m_{i,t,d}^2}{\sqrt{t}\hat{v}_{i,t,d}} + \frac{\alpha}{\sqrt{T(1-\beta_2)}}\frac{(\sum_{l=1}^{T}\beta_1^{T-l})(\sum_{l=1}^{T}\beta_1^{T-l}g_{i,l,d}^2)}{\sqrt{\sum_{l=1}^{T}\beta_2^{T-l}g_{i,l,d}^2}}$$

$$\stackrel{(iii)}{\leq} \sum_{t=1}^{T-1}\frac{\alpha m_{i,t,d}^2}{\sqrt{t}\hat{v}_{i,t,d}} + \frac{\alpha}{(1-\beta_1)\sqrt{T(1-\beta_2)}}\sum_{l=1}^{T}\frac{\beta_1^{T-l}g_{i,l,d}^2}{\sqrt{\beta_2^{T-l}g_{i,l,d}^2}}$$

$$\leq \sum_{t=1}^{T-1}\frac{\alpha m_{i,t,d}^2}{\sqrt{t}\hat{v}_{i,t,d}} + \frac{\alpha}{(1-\beta_1)\sqrt{T(1-\beta_2)}}\sum_{l=1}^{T}\eta^{T-l}|g_{i,l,d}|,$$

where (i) follows from the fact that the update rules of $m_T$ and $v_T$ can be written as $m_T = (1 - \beta_1) \sum_{l=1}^{T} \beta_1^{T-l} g_l$ and $v_T = (1 - \beta_2) \sum_{l=1}^{T} \beta_2^{T-l} g_l^2$, respectively. (ii) follows from Cauchy-Schwarz inequality and the fact that $0 \leq \beta_1 < 1$. Inequality (iii) follows since $\sum_{l=1}^{T} \beta_1^{T-l} \leq \frac{1}{1-\beta_1}$. Hence, we have

$$
\begin{aligned}
\sum_{t=1}^{T} \frac{\alpha m_{i,t,d}^2}{(1-\beta_1)\sqrt{t \hat{v}_{i,t,d}}} &\leq \sum_{t=1}^{T} \frac{\alpha}{(1-\beta_1)\sqrt{t(1-\beta_2)}} \sum_{l=1}^{t} \eta^{t-l} |g_{i,l,d}| \\
&= \frac{\alpha}{(1-\beta_1)\sqrt{(1-\beta_2)}} \sum_{t=1}^{T} \frac{1}{\sqrt{t}} \sum_{l=1}^{t} \eta^{t-l} |g_{i,l,d}| \\
&= \frac{\alpha}{(1-\beta_1)\sqrt{(1-\beta_2)}} \sum_{t=1}^{T} |g_{i,t,d}| \sum_{l=t}^{T} \frac{\eta^{l-t}}{\sqrt{l}} \\
&\leq \frac{\alpha}{(1-\beta_1)\sqrt{(1-\beta_2)}} \sum_{t=1}^{T} |g_{i,t,d}| \sum_{l=t}^{T} \frac{\eta^{l-t}}{\sqrt{t}} \\
&\overset{(i)}{\leq} \frac{\alpha}{(1-\beta_1)\sqrt{(1-\beta_2)}} \sum_{t=1}^{T} |g_{i,t,d}| \frac{1}{(1-\eta)\sqrt{t}} \\
&\overset{(ii)}{\leq} \frac{\alpha}{(1-\beta_1)(1-\eta)\sqrt{(1-\beta_2)}} \|g_{i,1:T,d}\| \sqrt{\sum_{t=1}^{T} \frac{1}{t}} \\
&\overset{(iii)}{\leq} \frac{\alpha\sqrt{1+\log T}}{(1-\beta_1)(1-\eta)\sqrt{(1-\beta_2)}} \|g_{i,1:T,d}\|,
\end{aligned}
$$

where inequality (i) follows since $\sum_{l=t}^{T} \eta^{l-t} \leq \frac{1}{(1-\eta)}$. Inequality (ii) follows from Cauchy-Schwarz inequality. The inequality (iii) follows since

$$
\sum_{t=1}^{T} \frac{1}{t} \leq 1 + \int_{t=1}^{T} \frac{1}{t} dt = 1 + \log t|_1^T = 1 + \log T. \tag{19}
$$

$\square$

Next, we provide an upper bound on the deviation of the local estimates at each iteration from their consensual value. A similar result has been proven in Shahrampour & Jadbabaie (2017) for online decentralized mirror descent; however, the following lemma extends that of Shahrampour & Jadbabaie (2017) to the online adaptive setting and takes into account the sparsity of gradient vector.

**Lemma 15.** ( *Network Error with Sparse Data) If $\beta_1, \beta_2 \in [0,1)$ satisfy $\eta = \frac{\beta_1}{\sqrt{\beta_2}} < 1$, then the sequence $x_{i,t}$ generated by Algorithm 1 satisfies*

$$
\sum_{t=1}^{T} \sum_{i=1}^{n} \frac{1}{\alpha_t} \|\hat{V}_{i,t}^{\frac{1}{4}}(\bar{x}_t - x_{i,t})\|^2 \leq \frac{n\sqrt{n}\alpha\sqrt{1+\log T} \sum_{d=1}^{p} \|g_{1:T,d}\|}{(1-\sigma_2(W))^2(1-\beta_1)(1-\eta)\sqrt{(1-\beta_2)}},
$$

*where $\hat{V}_{i,t} = diag(\hat{v}_{i,t})$ and $\bar{x}_t = \frac{1}{n}\sum_{i=1}^{n} x_{i,t}$.*

*Proof.* Let $e_{i,t} := x_{i,t+1} - \sum_{j=1}^{n}[W]_{ij}x_{j,t}$, where $W$ satisfies equation 2. Using the update rule of $x_{i,t+1}$ in Algorithm 1, we have

$$
\begin{aligned}
\sum_{t=1}^{T}\frac{1}{\alpha_t}\|\hat{V}_{i,t}^{\frac{1}{4}}e_{i,t}\|^2 &= \sum_{t=1}^{T}\frac{1}{\alpha_t}\|\hat{V}_{i,t}^{\frac{1}{4}}(x_{i,t+1} - \sum_{j=1}^{n}[W]_{ij}x_{j,t})\|^2 \\
&= \sum_{t=1}^{T}\frac{1}{\alpha_t}\|\hat{V}_{i,t}^{\frac{1}{4}}(\Pi_{\mathscr{X},\sqrt{\hat{v}_{i,t}}}[\sum_{j=1}^{n}[W]_{ij}x_{j,t} - \frac{\alpha_t m_{i,t}}{\sqrt{\hat{v}_{i,t}}}] - \sum_{j=1}^{n}[W]_{ij}x_{j,t})\|^2 \\
&\leq \sum_{t=1}^{T}\frac{1}{\alpha_t}\|\hat{V}_{i,t}^{\frac{1}{4}}(\sum_{j=1}^{n}[W]_{ij}x_{j,t} - \alpha_t\hat{V}_{i,t}^{\frac{-1}{2}}m_{i,t} - \sum_{j=1}^{n}[W]_{ij}x_{j,t})\|^2 \\
&= \sum_{t=1}^{T}\sum_{d=1}^{p}\frac{\alpha_t m_{i,t,d}^2}{\sqrt{\hat{v}_{i,t,d}}},
\end{aligned}
\tag{20}
$$

where the first inequality follows from Lemma 13.

Further, from the definition of $e_{i,t}$, we have

$$
x_{i,t+1} = \sum_{j=1}^{n}[W]_{ij}x_{j,t} + e_{i,t}.
\tag{21}
$$

Now, from equation 2 and equation 21, we have

$$
\begin{aligned}
\bar{x}_{t+1} = \frac{1}{n}\sum_{i=1}^{n}x_{i,t+1} &= \frac{1}{n}\sum_{i=1}^{n}\sum_{j=1}^{n}[W]_{ij}x_{j,t} + \frac{1}{n}\sum_{i=1}^{n}e_{i,t} \\
&= \frac{1}{n}\sum_{j=1}^{n}(\sum_{i=1}^{n}[W]_{ij})x_{j,t} + \frac{1}{n}\sum_{i=1}^{n}e_{i,t} \\
&= \bar{x}_t + \bar{e}_t,
\end{aligned}
$$

where $\bar{e}_t = \frac{1}{n}\sum_{i=1}^{n}e_{i,t}$. Hence,

$$
\bar{x}_{t+1} = \sum_{s=0}^{t}\bar{e}_s.
\tag{22}
$$

It follows from equation 21 that

$$
x_{i,t+1} = \sum_{s=0}^{t}\sum_{j=1}^{n}[W^{t-s}]_{ij}e_{i,s}.
\tag{23}
$$

Now, using equation 23 and equation 22, we have

$$
x_{i,t+1} - \bar{x}_{t+1} = \sum_{s=0}^{t}\sum_{j=1}^{n}([W^{t-s}]_{ij} - \frac{1}{n})e_{i,s}.
\tag{24}
$$

Now, taking the Euclidean norm of equation 24 and summing over $t \in \{1,\ldots,T\}$, one has:

$$
\begin{aligned}
\sum_{t=1}^{T}\frac{1}{\alpha_t}\|\hat{V}_{i,t}^{\frac{1}{4}}(x_{i,t+1} - \bar{x}_{t+1})\|^2 &\overset{(i)}{\leq} \sum_{t=1}^{T}t\sum_{s=0}^{t}(\sum_{j=1}^{n}|[W^{t-s}]_{ij} - \frac{1}{n}|)^2\frac{\|\hat{V}_{i,s}^{\frac{1}{4}}e_{i,s}\|^2}{\alpha_s} \\
&\overset{(ii)}{\leq} \sum_{s=0}^{T}\frac{\|\hat{V}_{i,s}^{\frac{1}{4}}e_{i,s}\|^2}{\alpha_s}\sum_{t=1}^{T}nt\sigma_2^{2t-2s}(W) \\
&\overset{(iii)}{\leq} \frac{n}{(1-\sigma_2(W))^2}\sum_{t=0}^{T}\frac{\|\hat{V}_{i,t}^{\frac{1}{4}}e_{i,t}\|^2}{\alpha_t} \\
&\overset{(iv)}{\leq} \frac{n}{(1-\sigma_2(W))^2}\sum_{d=1}^{p}\sum_{t=1}^{T}\frac{\alpha_t m_{i,t,d}^2}{\sqrt{\hat{v}_{i,t,d}}} \\
&\overset{(v)}{\leq} \frac{n\alpha\sqrt{1+\log T}\sum_{d=1}^{p}\|g_{i,1:T,d}\|}{(1-\sigma_2(W))^2(1-\beta_1)(1-\eta)\sqrt{(1-\beta_2)}},
\end{aligned}
\tag{25}
$$

where step (i) follows from $\|\sum_{i=1}^{n} a_i\|^2 \le n \sum_{i=1}^{n} \|a_i\|^2$, step (ii) follows from the following property of mixing matrix $W$ Horn et al. (1990),

$$\sum_{j=1}^{n} \left| [W^t]_{ij} - \frac{1}{n} \right| \le \sqrt{n} \sigma_2^t(W), \tag{26}$$

step (iii) follows from $\sum_{t=1}^{T} t\sigma_2^t(W) < \frac{1}{(1-\sigma_2(W))^2}$, step (iv) follows from equation 20 and step (v) follows from Lemma 14.

Now, summing the equation 25 over $i \in \mathscr{V}$ and using

$$\sum_{i=1}^{n} \|g_{i,1:T,d}\| \le \sqrt{n} (\sum_{i=1}^{n} \|g_{i,1:T,d}\|^2)^{\frac{1}{2}} = \sqrt{n} \|g_{1:T,d}\|, \tag{27}$$

we complete the proof. $\qquad\square$

**Lemma 16.** *For the sequence $x_{i,t}$ generated by Algorithm 1, we have*

$$\frac{1}{n} \sum_{i=1}^{n} \sum_{t=1}^{T} \left( \frac{\sqrt{\hat{v}_{i,t}}}{\alpha_t} \|x_t^* - \sum_{j=1}^{n} [W]_{ij} x_{j,t}\|^2 - \frac{\sqrt{\hat{v}_{i,t}}}{\alpha_t} \|x_t^* - x_{i,t+1}\|^2 \right)$$

$$\le \frac{2\gamma_\infty^2}{\sqrt{n}} \sum_{d=1}^{p} \frac{\sqrt{\hat{v}_{T,d}}}{\alpha_T} + \frac{2\gamma_\infty}{\sqrt{n}} \sum_{d=1}^{p} \frac{\sqrt{\hat{v}_{T,d}}}{\alpha_T} \sum_{t=1}^{T-1} |x_{t+1,d}^* - x_{t,d}^*|. \tag{28}$$

*Proof.* From the left side of equation 28, we have

$$\frac{\sqrt{\hat{v}_{i,t}}}{\alpha_t} \|x_t^* - \sum_{j=1}^{n} [W]_{ij} x_{j,t}\|^2 - \frac{\sqrt{\hat{v}_{i,t}}}{\alpha_t} \|x_t^* - x_{i,t+1}\|^2$$

$$= \frac{\sqrt{\hat{v}_{i,t}}}{\alpha_t} \|x_t^* - \sum_{j=1}^{n} [W]_{ij} x_{j,t}\|^2 - \frac{\sqrt{\hat{v}_{i,t+1}}}{\alpha_{t+1}} \|x_{t+1}^* - \sum_{j=1}^{n} [W]_{ij} x_{j,t+1}\|^2$$

$$+ \frac{\sqrt{\hat{v}_{i,t+1}}}{\alpha_{t+1}} \|x_{t+1}^* - \sum_{j=1}^{n} [W]_{ij} x_{j,t+1}\|^2 - \frac{\sqrt{\hat{v}_{i,t+1}}}{\alpha_{t+1}} \|x_t^* - \sum_{j=1}^{n} [W]_{ij} x_{j,t+1}\|^2 \tag{29}$$

$$+ \frac{\sqrt{\hat{v}_{i,t+1}}}{\alpha_{t+1}} \|x_t^* - x_{i,t+1}\|^2 - \frac{\sqrt{\hat{v}_{i,t}}}{\alpha_t} \|x_t^* - x_{i,t+1}\|^2. \tag{30}$$

By construction of equation 29, we have

$$\sqrt{\hat{v}_{i,t+1}} \|x_{t+1}^* - \sum_{j=1}^{n} [W]_{ij} x_{j,t+1}\|^2 - \sqrt{\hat{v}_{i,t+1}} \|x_t^* - \sum_{j=1}^{n} [W]_{ij} x_{j,t+1}\|^2$$

$$= \sum_{d=1}^{p} \sqrt{\hat{v}_{i,t+1,d}} \langle x_{t+1,d}^* - x_{t,d}^*, x_{t+1,d}^* + x_{t,d}^* - 2\sum_{j=1}^{n} [W]_{ij} x_{j,t+1,d} \rangle$$

$$\le 2\gamma_\infty \sum_{d=1}^{p} \sqrt{\hat{v}_{i,t+1,d}} |x_{t+1,d}^* - x_{t,d}^*|, \tag{31}$$

where the last inequality holds due to equation 1.

Summing equation 30 over $t \in \{1, \ldots, T\}$, the first term telescopes, while the second is handled with equation 31. Hence,

$$\sum_{t=1}^{T} \left( \frac{\sqrt{\hat{v}_{i,t}}}{\alpha_t} \|x_t^* - \sum_{j=1}^{n} [W]_{ij} x_{j,t}\|^2 - \frac{\sqrt{\hat{v}_{i,t}}}{\alpha_t} \|x_t^* - x_{i,t+1}\|^2 \right)$$

$$\le \sum_{d=1}^{p} \frac{\gamma_\infty^2 \sqrt{\hat{v}_{i,1,d}}}{\alpha_1} + \sum_{t=1}^{T-1} \sum_{d=1}^{p} \frac{2\gamma_\infty \sqrt{\hat{v}_{i,t+1,d}}}{\alpha_{t+1}} |x_{t+1,d}^* - x_{t,d}^*| + \gamma_\infty^2 \sum_{t=1}^{T-1} \sum_{d=1}^{p} \left( \frac{\sqrt{\hat{v}_{i,t+1,d}}}{\alpha_{t+1}} - \frac{\sqrt{\hat{v}_{i,t,d}}}{\alpha_t} \right)$$

$$\le \sum_{d=1}^{p} \frac{2\gamma_\infty^2 \sqrt{\hat{v}_{i,T,d}}}{\alpha_T} + \sum_{t=1}^{T-1} \sum_{d=1}^{p} \frac{2\gamma_\infty \sqrt{\hat{v}_{i,t+1,d}}}{\alpha_{t+1}} |x_{t+1,d}^* - x_{t,d}^*|, \tag{32}$$

where the last inequality follows since from definition of $\hat{v}_{i,t}$, we have

$$\frac{\sqrt{\hat{v}_{i,t+1,d}}}{\alpha_{t+1}} \geq \frac{\sqrt{\hat{v}_{i,t,d}}}{\alpha_t}. \tag{33}$$

Now, summing equation 32 over $i \in \mathcal{V}$ and using the inequality $\sum_{i=1}^n \sqrt{\hat{v}_{i,t}} \leq \sqrt{n}\sqrt{\hat{v}_t}$, the claim in equation 28 follows. □

**Lemma 17.** *Suppose that the parameters* $\beta_1, \beta_2 \in [0, 1)$ *satisfy* $\eta = \frac{\beta_1}{\sqrt{\beta_2}} < 1$. *Let* $\beta_{1,t} = \beta_1 \lambda^{t-1}, \lambda \in (0, 1)$ *and* $\|\nabla f_{i,t}(x)\|_\infty \leq G_\infty$ *for all* $t \in \{1, \dots, T\}$. *Then, using a step-size* $\alpha_t = \frac{\alpha}{\sqrt{t}}$ *for the sequence* $x_{i,t}$ *generated by Algorithm 1, we have*

$$\frac{1}{n}\sum_{i=1}^n \sum_{t=1}^T \left( f_{i,t}(x_{i,t}) - f_{i,t}(x_t^*) \right) \leq \frac{(1-\beta_1)\alpha\sqrt{1+\log T}}{2\sqrt{(1-\beta_2)}} \sum_{d=1}^p \|g_{1:T,d}\| + \sum_{d=1}^p \frac{G_\infty \gamma_\infty}{(1-\beta_1)(1-\lambda)}$$

$$+ \frac{2\alpha\sqrt{1+\log T}\sum_{d=1}^p \|g_{1:T,d}\|}{(1-\sigma_2(W))\sqrt{(1-\beta_1)}\sqrt{(1-\eta)}\sqrt{(1-\beta_2)}}$$

$$+ \frac{\gamma_\infty^2}{\sqrt{n}} \sum_{d=1}^p \frac{\sqrt{T\hat{v}_{T,d}}}{(1-\beta_1)\alpha} + \frac{\gamma_\infty}{\sqrt{n}} \sum_{d=1}^p \frac{\sqrt{T\hat{v}_{T,d}}}{(1-\beta_1)\alpha} \sum_{t=1}^{T-1} |x_{t+1,d}^* - x_{t,d}^*|.$$

*Proof.* From convexity of $f_{i,t}(\cdot)$, we have

$$\frac{1}{n}\sum_{i=1}^n \sum_{t=1}^T f_{i,t}(x_{i,t}) - f_{i,t}(x_t^*) \leq \frac{1}{n}\sum_{i=1}^n \sum_{t=1}^T \langle \nabla f_{i,t}(x_{i,t}), x_{i,t} - x_t^* \rangle$$

$$= \frac{1}{n}\sum_{i=1}^n \sum_{t=1}^T \underbrace{\langle \nabla f_{i,t}(x_{i,t}), x_{i,t+1} - x_t^* \rangle}_{I_1} + \frac{1}{n}\sum_{i=1}^n \sum_{t=1}^T \underbrace{\langle \nabla f_{i,t}(x_{i,t}), x_{i,t} - \sum_{j=1}^n [W]_{ij}x_{j,t} \rangle}_{I_2}$$

$$+ \frac{1}{n}\sum_{i=1}^n \sum_{t=1}^T \underbrace{\langle \nabla f_{i,t}(x_{i,t}), \sum_{j=1}^n [W]_{ij}x_{j,t} - x_{i,t+1} \rangle}_{I_3}. \tag{34}$$

Individual terms in equation 34 can be bounded in the following way. From the Young's inequality for products[3], we have

$$I_3 = \frac{1}{n}\sum_{i=1}^n \sum_{t=1}^T \langle \nabla f_{i,t}(x_{i,t}), \sum_{j=1}^n [W]_{ij}x_{j,t} - x_{i,t+1} \rangle$$

$$\leq \frac{1}{n}\sum_{i=1}^n \sum_{t=1}^T \frac{\sqrt{\hat{v}_{i,t}}}{2\alpha_t(1-\beta_1)} \|\sum_{j=1}^n [W]_{ij}x_{j,t} - x_{i,t+1}\|^2 + \frac{1}{n}\sum_{i=1}^n \sum_{t=1}^T \frac{(1-\beta_1)\alpha_t}{2\sqrt{\hat{v}_{i,t}}} \|\nabla f_{i,t}(x_{i,t})\|^2. \tag{35}$$

Note also that:

$$\sum_{t=1}^T \frac{\alpha_t g_{i,t,d}^2}{\sqrt{\hat{v}_{i,t,d}}} = \sum_{t=1}^{T-1} \frac{\alpha_t g_{i,t,d}^2}{\sqrt{\hat{v}_{i,t,d}}} + \frac{\alpha_T g_{i,T,d}^2}{\sqrt{\hat{v}_{i,T,d}}}$$

$$= \sum_{t=1}^{T-1} \frac{\alpha_t g_{i,t,d}^2}{\sqrt{\hat{v}_{i,t,d}}} + \frac{\alpha g_{i,T,d}^2}{\sqrt{T(1-\beta_2)}\sqrt{\sum_{l=1}^T \beta_2^{T-l}g_{i,l,d}^2}}$$

$$\leq \sum_{t=1}^{T-1} \frac{\alpha_t g_{i,t,d}^2}{\sqrt{\hat{v}_{i,t,d}}} + \frac{\alpha |g_{i,T,d}|}{\sqrt{T(1-\beta_2)}}.$$

---

[3]An elementary case of Young's inequality is $ab \leq \frac{a^2}{2} + \frac{b^2}{2}$.

Using Cauchy-Schwarz inequality, equation 19 and equation 27, we have

$$
\sum_{i=1}^{n} \sum_{t=1}^{T} \frac{\alpha_t g_{i,t,d}^2}{\sqrt{\hat{v}_{i,t,d}}} \leq \frac{\alpha}{\sqrt{(1-\beta_2)}} \sum_{i=1}^{n} \sum_{t=1}^{T} \frac{|g_{i,t,d}|}{\sqrt{t}}
$$

$$
\leq \frac{\alpha}{\sqrt{(1-\beta_2)}} \sum_{i=1}^{n} \|g_{i,1:T,d}\| \sqrt{\sum_{t=1}^{T} \frac{1}{t}}
$$

$$
\leq \frac{\alpha\sqrt{1+\log T}}{\sqrt{(1-\beta_2)}} \sum_{i=1}^{n} \|g_{i,1:T,d}\|
$$

$$
\leq \frac{\sqrt{n}\alpha\sqrt{1+\log T}}{\sqrt{(1-\beta_2)}} \|g_{1:T,d}\|. \tag{36}
$$

Using equation 35 and equation 36, we have

$$
I3 \leq \frac{1}{n} \sum_{i=1}^{n} \sum_{t=1}^{T} \frac{\sqrt{\hat{v}_{i,t}}}{2\alpha_t(1-\beta_1)} \| \sum_{j=1}^{n} [W]_{ij} x_{j,t} - z_{i,t+1}\|^2 + \frac{(1-\beta_1)\sqrt{n}\alpha\sqrt{1+\log T}}{2\sqrt{(1-\beta_2)}} \|g_{1:T,d}\|. \tag{37}
$$

In addition, we have

$$
I_2 = \frac{1}{n} \sum_{i=1}^{n} \sum_{t=1}^{T} \langle g_{i,t}, x_{i,t} - \sum_{j=1}^{n} [W]_{ij} x_{j,t} \rangle
$$

$$
= \frac{1}{n} \sum_{i=1}^{n} \sum_{t=1}^{T} \langle g_{i,t}, x_{i,t} - \bar{x}_t + \bar{x}_t - \sum_{j=1}^{n} [W]_{ij} x_{j,t} \rangle
$$

$$
= \frac{1}{n} \sum_{i=1}^{n} \sum_{t=1}^{T} \langle g_{i,t}, x_{i,t} - \bar{x}_t \rangle + \frac{1}{n} \sum_{i=1}^{n} \sum_{t=1}^{T} \sum_{j=1}^{n} [W]_{ij} \langle g_{i,t}, \bar{x}_t - x_{j,t} \rangle
$$

$$
= \frac{1}{n} \sum_{i=1}^{n} \sum_{t=1}^{T} \langle \sqrt{\alpha_t} \hat{V}_{i,t}^{\frac{-1}{4}} g_{i,t}, \frac{\hat{V}_{i,t}^{\frac{1}{4}}}{\sqrt{\alpha_t}} (x_{i,t} - \bar{x}_t) \rangle
$$

$$
+ \frac{1}{n} \sum_{i=1}^{n} \sum_{t=1}^{T} \sum_{j=1}^{n} [W]_{ij} \langle \sqrt{\alpha_t} \hat{V}_{i,t}^{\frac{-1}{4}} g_{i,t}, \frac{\hat{V}_{i,t}^{\frac{1}{4}}}{\sqrt{\alpha_t}} (\bar{x}_t - x_{j,t}) \rangle
$$

$$
\leq \frac{1}{n} \sum_{i=1}^{n} \sqrt{\sum_{t=1}^{T} \alpha_t \hat{V}_{i,t}^{\frac{-1}{2}} g_{i,t}^2} \sqrt{\sum_{t=1}^{T} \frac{\hat{V}_{i,t}^{\frac{1}{2}}}{\alpha_t} (x_{i,t} - \bar{x}_t)^2}
$$

$$
+ \frac{1}{n} \sum_{i=1}^{n} \sum_{j=1}^{n} [W]_{ij} \sqrt{\sum_{t=1}^{T} \alpha_t \hat{V}_{i,t}^{\frac{-1}{2}} g_{i,t}^2} \sqrt{\sum_{t=1}^{T} \frac{\hat{V}_{i,t}^{\frac{1}{2}}}{\alpha_t} (x_{j,t} - \bar{x}_t)^2} \tag{38}
$$

where equation 38 follows from Cauchy-Schwarz inequality.

Now, using equation 38 we obtain

$$
I_2 \leq \frac{1}{n} \sqrt{\sum_{i=1}^{n} \sum_{t=1}^{T} \sum_{d=1}^{p} \alpha_t \hat{v}_{i,t,d}^{\frac{-1}{2}} g_{i,t,d}^2} \sqrt{\sum_{i=1}^{n} \sum_{t=1}^{T} \sum_{d=1}^{p} \frac{\hat{v}_{i,t,d}^{\frac{1}{2}}}{\alpha_t} (x_{i,t,d} - \bar{x}_{t,d})^2}
$$

$$
+ \frac{1}{n} \sum_{j=1}^{n} [W]_{ij} \sqrt{\sum_{i=1}^{n} \sum_{t=1}^{T} \sum_{d=1}^{p} \alpha_t \hat{v}_{i,t,d}^{\frac{-1}{2}} g_{i,t,d}^2} \sqrt{\sum_{i=1}^{n} \sum_{t=1}^{T} \sum_{d=1}^{p} \frac{\hat{v}_{i,t,d}^{\frac{1}{2}}}{\alpha_t} (x_{j,t,d} - \bar{x}_{t,d})^2} \tag{39}
$$

$$
\leq \frac{2\alpha\sqrt{1+\log T} \sum_{d=1}^{p} \|g_{1:T,d}\|}{(1-\sigma_2(W))\sqrt{(1-\beta_1)}\sqrt{(1-\eta)}\sqrt{(1-\beta_2)}}, \tag{40}
$$

where equation 39 utilize Cauchy-Schwarz inequality and equation 40 follows from equation 2, Lemma 15 and equation 36.

To bound $I_1$, using the update rule of $\hat{m}_{i,t}$ in Algorithm 1, we have

$$\langle \alpha_t \frac{m_{i,t}}{\sqrt{\hat{v}_{i,t}}}, x_{i,t+1} - x_t^* \rangle = \langle \alpha_t (\frac{\beta_{1,t}}{\sqrt{\hat{v}_{i,t}}} m_{i,t-1} + \frac{(1-\beta_{1,t})}{\sqrt{\hat{v}_{i,t}}} \nabla f_{i,t}(x_{i,t})), x_{i,t+1} - x_t^* \rangle$$

$$= \langle \frac{\alpha_t \beta_{1,t}}{\sqrt{\hat{v}_{i,t}}} m_{i,t-1}, x_{i,t+1} - x_t^* \rangle + \langle \frac{\alpha_t (1-\beta_{1,t})}{\sqrt{\hat{v}_{i,t}}} \nabla f_{i,t}(x_{i,t}), x_{i,t+1} - x_t^* \rangle.$$

Now, by rearranging the above equality, we obtain:

$$\sum_{d=1}^{p} \langle \frac{(1-\beta_{1,t})}{\sqrt{\hat{v}_{i,t,d}}} \nabla f_{i,t,d}(x_{i,t,d}), x_{i,t+1,d} - x_{t,d}^* \rangle$$

$$= \sum_{d=1}^{p} \langle \frac{m_{i,t,d}}{\sqrt{\hat{v}_{i,t,d}}}, x_{i,t+1,d} - x_{t,d}^* \rangle + \sum_{d=1}^{p} \frac{\beta_{1,t}}{\sqrt{\hat{v}_{i,t,d}}} \langle m_{i,t-1,d}, x_{t,d}^* - x_{i,t+1,d} \rangle$$

$$\leq \langle \frac{m_{i,t}}{\sqrt{\hat{v}_{i,t}}}, x_{i,t+1} - x_t^* \rangle + ||x_t^* - x_{i,t+1}||_\infty \sum_{d=1}^{p} \frac{\beta_{1,t} m_{i,t-1,d}}{\sqrt{\hat{v}_{i,t,d}}}$$

$$\leq \underbrace{\frac{1}{2\alpha_t} ||x_t^* - \sum_{j=1}^{n} [W]_{ij} x_{j,t}||^2 - \frac{1}{2\alpha_t} ||x_t^* - x_{i,t+1}||^2 - \frac{1}{2\alpha_t} ||x_{i,t+1} - \sum_{j=1}^{n} [W]_{ij} x_{j,t}||^2}_{(II_1)}$$

$$+ \underbrace{||x_t^* - x_{i,t+1}||_\infty G_\infty \sum_{d=1}^{p} \frac{\beta_{1,t}}{\sqrt{\hat{v}_{i,t,d}}}}_{(II_2)}, \tag{41}$$

where $(II_1)$ follows from Lemma 12. The term $(II_2)$ is obtained by induction as follows: using the assumption, $||g_{i,t}||_\infty \leq G_\infty$; now, using the update rule of $m_{i,t}$ in Algorithm 1, we have

$$||m_{i,t}||_\infty \leq (\beta_1 + (1-\beta_1)) \max(||g_{i,t}||_\infty, ||m_{i,t-1}||_\infty) = \max(||g_{i,t}||_\infty, ||m_{i,t-1}||_\infty) \leq G_\infty, \tag{42}$$

where $||m_{i,t-1}||_\infty \leq G_\infty$ by induction hypothesis.

Note that from our assumption, we have $\beta_{1,t} = \beta_1 \lambda^{t-1}, \lambda \in (0,1)$ and $\beta_{1,t} \leq \beta_1$. Substituting this into equation 41, we get

$$I_1 = \langle \nabla f_{i,t}(x_{i,t}), x_{i,t+1} - x_t^* \rangle$$

$$\leq \frac{\sqrt{\hat{v}_{i,t}}}{(1-\beta_{1,t})} [\frac{1}{2\alpha_t} ||x_t^* - \sum_{j=1}^{n} [W]_{ij} x_{j,t}||^2 - \frac{1}{2\alpha_t} ||x_t^* - x_{i,t+1}||^2$$

$$- \frac{1}{2\alpha_t} ||x_{i,t+1} - \sum_{j=1}^{n} [W]_{ij} x_{j,t}||^2] + ||x_t^* - x_{i,t+1}||_\infty G_\infty \sum_{d=1}^{p} \frac{\beta_{1,t}}{(1-\beta_{1,t})}$$

$$\leq \frac{\sqrt{\hat{v}_{i,t}}}{(1-\beta_1)} [\frac{1}{2\alpha_t} ||x_t^* - \sum_{j=1}^{n} [W]_{ij} x_{j,t}||^2 - \frac{1}{2\alpha_t} ||x_t^* - x_{i,t+1}||^2]$$

$$- \frac{\sqrt{\hat{v}_{i,t}}}{(1-\beta_1) 2\alpha_t} ||x_{i,t+1} - \sum_{j=1}^{n} [W]_{ij} x_{j,t}||^2 + \gamma_\infty G_\infty \sum_{d=1}^{p} \frac{\beta_{1,t}}{(1-\beta_{1,t})}, \tag{43}$$

where the last line comes from equation 1.

Plugging equation 37, equation 40, and equation 43 into equation 34, we obtain

$$\frac{1}{n} \sum_{i=1}^{n} \sum_{t=1}^{T} \left( f_{i,t}(x_{i,t}) - f_{i,t}(x_t^*) \right) \leq \frac{(1-\beta_1)\alpha \sqrt{1+\log T}}{2\sqrt{n}\sqrt{(1-\beta_2)}} \sum_{d=1}^{p} ||g_{1:T,d}|| + \gamma_\infty G_\infty \sum_{t=1}^{T} \sum_{d=1}^{p} \frac{\beta_{1,t}}{(1-\beta_{1,t})}$$

$$+ \frac{2\alpha \sqrt{1+\log T} \sum_{d=1}^{p} ||g_{1:T,d}||}{(1-\sigma_2(W))\sqrt{(1-\beta_1)}\sqrt{(1-\eta)}\sqrt{(1-\beta_2)}}$$

$$+ \frac{1}{n} \sum_{i=1}^{n} \sum_{t=1}^{T} \frac{\sqrt{\hat{v}_{i,t}}}{2(1-\beta_1)} [\frac{1}{\alpha_t} ||x_t^* - \sum_{j=1}^{n} [W]_{ij} x_{j,t}||^2 - \frac{1}{\alpha_t} ||x_t^* - x_{i,t+1}||^2].$$

Now, since $\beta_{1,t} = \beta_1 \lambda^{t-1}, \lambda \in (0,1)$ and $\beta_{1,t} \leq \beta_1$, we have

$$\sum_{t=1}^{T} \frac{\beta_{1,t}}{(1-\beta_{1,t})} \leq \sum_{t=1}^{T} \frac{\lambda^{t-1}}{(1-\beta_1)} \leq \frac{1}{(1-\beta_1)(1-\lambda)}. \tag{44}$$

Finally, using Lemma 16 and equation 44, we obtain the desired result. $\square$

### 6.1.1 PROOF OF THEOREM 4

*Proof.* From convexity of $f_{i,t}(\cdot)$ it follows that

$$\begin{aligned} f_t(x_{i,t}) - f_t(x_t^*) &= f_t(x_{i,t}) - f_t(\bar{x}_t) + f_t(\bar{x}_t) - f_t(x_t^*) \\ &\leq \langle g_{i,t}, x_{i,t} - \bar{x}_t \rangle + f_t(\bar{x}_t) - f_t(x_t^*) \\ &= \frac{1}{n} \sum_{i=1}^{n} f_{i,t}(\bar{x}_t) - \frac{1}{n} \sum_{i=1}^{n} f_{i,t}(x_t^*) + \langle g_{i,t}, x_{i,t} - \bar{x}_t \rangle, \end{aligned}$$

and

$$f_t(x_{i,t}) - f_t(x_t^*) \leq \frac{1}{n} \sum_{i=1}^{n} f_{i,t}(x_{i,t}) - \frac{1}{n} \sum_{i=1}^{n} f_{i,t}(x_t^*) + \langle g_{i,t}, x_{i,t} - \bar{x}_t \rangle + \frac{1}{n} \sum_{i=1}^{n} \langle g_{i,t}, x_{i,t} - \bar{x}_t \rangle. \tag{45}$$

Summing over $t \in \{1, \dots, T\}$ and $i \in \mathcal{V}$, and applying Lemma 17 and equation 40 gives the desired result. $\square$

### 6.1.2 PROOF OF THEOREM 5

*Proof.* We just need to rework the proof of Lemma 17 and Theorem 4 using stochastic gradients by tracking the changes. Indeed, using stochastic gradient at the beginning of Lemma 17, we have

$$\begin{aligned} \sum_{t=1}^{T} \left( f_{i,t}(x_{i,t}) - f_{i,t}(x_t^*) \right) &\leq \sum_{t=1}^{T} \langle \nabla f_{i,t}(x_{i,t}), x_{i,t} - x_t^* \rangle \\ &= \sum_{t=1}^{T} \langle \boldsymbol{\nabla} f_{i,t}(x_{i,t}), x_{i,t} - x_t^* \rangle + \sum_{t=1}^{T} \langle \nabla f_{i,t}(x_{i,t}) - \boldsymbol{\nabla} f_{i,t}(x_{i,t}), x_{i,t} - x_t^* \rangle \\ &= \sum_{t=1}^{T} \langle \boldsymbol{\nabla} f_{i,t}(x_{i,t}), x_{i,t+1} - x_t^* \rangle + \sum_{t=1}^{T} \langle \boldsymbol{\nabla} f_{i,t}(x_{i,t}), x_{i,t} - \sum_{j=1}^{n} [W]_{ij} x_{j,t} \rangle \\ &\quad + \sum_{t=1}^{T} \langle \boldsymbol{\nabla} f_{i,t}(x_{i,t}), \sum_{j=1}^{n} [W]_{ij} x_{j,t} - x_{i,t+1} \rangle + \sum_{t=1}^{T} \langle \nabla f_{i,t}(x_{i,t}) - \boldsymbol{\nabla} f_{i,t}(x_{i,t}), x_{i,t} - x_t^* \rangle. \end{aligned}$$

Further, if we replace any bound involving $G_\infty$ which is an upper bound on the exact gradient with the norm of stochastic gradient, we obtain

$$\begin{aligned} \frac{1}{n} \sum_{i=1}^{n} \sum_{t=1}^{T} \left( f_{i,t}(x_{i,t}) - f_{i,t}(x_t^*) \right) &\leq \frac{(1-\beta_1)\alpha\sqrt{1+\log T}}{2\sqrt{n}\sqrt{(1-\beta_2)}} \sum_{d=1}^{p} \|g_{1:T,d}\| + \gamma_\infty \|m_{i,t-1}\|_\infty \sum_{t=1}^{T} \sum_{d=1}^{p} \frac{\beta_{1,t}}{(1-\beta_{1,t})} \\ &\quad + \frac{2\alpha\sqrt{1+\log T} \sum_{d=1}^{p} \|g_{1:T,d}\|}{(1-\sigma_2(W))\sqrt{(1-\beta_1)}\sqrt{(1-\eta)}\sqrt{(1-\beta_2)}} \\ &\quad + \frac{1}{n} \sum_{i=1}^{n} \sum_{t=1}^{T} \frac{\sqrt{\hat{v}_{i,t}}}{2(1-\beta_1)} \left[ \frac{1}{\alpha_t} \|x_t^* - \sum_{j=1}^{n} [W]_{ij} x_{j,t}\|^2 - \frac{1}{\alpha_t} \|x_t^* - x_{i,t+1}\|^2 \right] \\ &\quad + \frac{1}{n} \sum_{i=1}^{n} \langle \nabla f_{i,t}(x_{i,t}) - \boldsymbol{\nabla} f_{i,t}(x_{i,t}), x_{i,t} - x_t^* \rangle. \end{aligned} \tag{46}$$

Note that using Assumption 3, we have

$$\mathbb{E}\left[ \langle \nabla f_{i,t}(x_{i,t}) - \boldsymbol{\nabla} f_{i,t}(x_{i,t}), x_{i,t} - x_t^* \rangle \right] = 0,$$

and

$$\mathbb{E}\left[ \|\boldsymbol{\nabla} f_{i,t}(x_{i,t})\| \right] \leq \left( \mathbb{E}\left[ \|\boldsymbol{\nabla} f_{i,t}(x_{i,t})\|^2 \right] \right)^{\frac{1}{2}} \leq \xi.$$

Hence taking expectation from equation 46, using the above inequality and equation 42, we have

$$\frac{1}{n}\sum_{i=1}^{n}\sum_{t=1}^{T}\left(\mathbb{E}\left[f_{i,t}(x_{i,t})\right]-f_{i,t}(x_t^*)\right) \leq \frac{(1-\beta_1)\alpha\sqrt{1+\log T}}{2\sqrt{n}\sqrt{(1-\beta_2)}}\sum_{d=1}^{p}\mathbb{E}\left[\|g_{1:T,d}\|\right]+\gamma_\infty\xi\sum_{t=1}^{T}\sum_{d=1}^{p}\frac{\beta_{1,t}}{(1-\beta_{1,t})}$$
$$+\frac{2\alpha\sqrt{1+\log T}\sum_{d=1}^{p}\mathbb{E}\left[\|g_{1:T,d}\|\right]}{(1-\sigma_2(W))\sqrt{(1-\beta_1)}\sqrt{(1-\eta)}\sqrt{(1-\beta_2)}}$$
$$+\frac{1}{n}\sum_{i=1}^{n}\frac{1}{2(1-\beta_1)}\mathbb{E}\left[\frac{\sqrt{\hat{v}_{i,t}}}{\alpha_t}\|x_t^*-\sum_{j=1}^{n}[W]_{ij}x_{j,t}\|^2-\frac{\sqrt{\hat{v}_{i,t}}}{\alpha_t}\|x_t^*-x_{i,t+1}\|^2\right].$$

According to the result in Lemma 16 and equation 44, we have

$$\frac{1}{n}\sum_{i=1}^{n}\sum_{t=1}^{T}\left(\mathbb{E}\left[f_{i,t}(x_{i,t})\right]-f_{i,t}(x_t^*)\right) \leq \frac{(1-\beta_1)\alpha\sqrt{1+\log T}}{2\sqrt{n}\sqrt{(1-\beta_2)}}\sum_{d=1}^{p}\mathbb{E}\left[\|g_{1:T,d}\|\right]+\sum_{d=1}^{p}\frac{\xi\gamma_\infty}{(1-\beta_1)(1-\lambda)}$$
$$+\frac{2\alpha\sqrt{1+\log T}\sum_{d=1}^{p}\mathbb{E}\left[\|g_{1:T,d}\|\right]}{(1-\sigma_2(W))\sqrt{(1-\beta_1)}\sqrt{(1-\eta)}\sqrt{(1-\beta_2)}}$$
$$+\frac{\gamma_\infty^2}{\sqrt{n}}\sum_{d=1}^{p}\frac{\sqrt{T}\mathbb{E}\left[\sqrt{\hat{v}_{T,d}}\right]}{(1-\beta_1)\alpha}+\frac{\gamma_\infty}{\sqrt{n}}\sum_{d=1}^{p}\frac{\sqrt{T}\mathbb{E}\left[\sqrt{\hat{v}_{T,d}}\right]}{(1-\beta_1)\alpha}\sum_{t=1}^{T-1}|x_{t+1,d}^*-x_{t,d}^*|.$$

Finally, using equation 45 we complete the proof. □

Next, we establish a series of lemmas used in the proof of Theorems 8 and 10.

**Lemma 18.** *Attouch et al. (2013) Let $G_{\mathscr{X}}$ be the projected gradient defied in equation 5. Then, $G_{\mathscr{X}}(\varpi, f_i, \alpha) = 0$ if and only if $\varpi$ is a critical point of equation 3.*

Next, we show that the projection map $G_{\mathscr{X}}$ in Definition 1 is Lipschitz continuous.

**Lemma 19.** *Suppose the second moment $\hat{v}$ satisfies equation 11. Let $x_{1,i}^+$ and $x_{2,i}^+$ be given in Definition 1 with $\frac{m_i}{\sqrt{\hat{v}_i}}$ replaced by $\frac{m_i^1}{\sqrt{\hat{v}_i^1}}$ and $\frac{m_i^2}{\sqrt{\hat{v}_i^2}}$, respectively. Then,*

$$\|G_{\mathscr{X}}^1(x_i, \bar{f}_i, \alpha) - G_{\mathscr{X}}^2(x_i, \bar{f}_i, \alpha)\| \leq \frac{\bar{v}}{\underline{v}}\|m_i^1 - m_i^2\|, \qquad \forall i \in \mathscr{V},$$

*where $G_{\mathscr{X}}^1$ and $G_{\mathscr{X}}^2$ are the projection maps corresponding to $x_{1,i}^+$ and $x_{2,i}^+$, respectively.*

*Proof.* Consider the optimality condition of equation 6, for any $u \in \mathscr{X}$. For each $i \in \mathscr{V}$, observe that

$$\langle\frac{m_i^1}{\sqrt{\hat{v}_i^1}}+\frac{1}{\alpha}(x_{1,i}^+-\sum_{j=1}^{n}[W]_{ij}x_j), u-x_{1,i}^+\rangle \geq 0, \tag{47}$$

$$\langle\frac{m_i^2}{\sqrt{\hat{v}_i^2}}+\frac{1}{\alpha}(x_{2,i}^+-\sum_{j=1}^{n}[W]_{ij}x_j), u-x_{2,i}^+\rangle \geq 0. \tag{48}$$

Taking $u = x_{2,i}^+$ in equation 47, we have

$$\langle m_i^1, x_{2,i}^+-x_{1,i}^+\rangle \geq \frac{\sqrt{\hat{v}_i^1}}{\alpha}\langle\sum_{j=1}^{n}[W]_{ij}x_j-x_{1,i}^+, x_{2,i}^+-x_{1,i}^+\rangle. \tag{49}$$

Likewise, setting $u = x_{1,i}^+$ in equation 48, we get

$$\langle m_i^2, x_{1,i}^+-x_{2,i}^+\rangle \geq \frac{\sqrt{\hat{v}_i^2}}{\alpha}\langle\sum_{j=1}^{n}[W]_{ij}x_j-x_{2,i}^+, x_{1,i}^+-x_{2,i}^+\rangle. \tag{50}$$

Now, using equation 49, equation 50 and equation 11, we obtain

$$\|m_i^1 - m_i^2\| \geq \frac{\underline{v}}{\alpha}\|x_{2,i}^+-x_{1,i}^+\|. \tag{51}$$

Without loss of generality, assumes that $\sqrt{\hat{v}_i^1} \leq \sqrt{\hat{v}_i^2}$. Then, using equation 5, we have

$$\|G_{\mathscr{X}}^1(x_i, \bar{f}_i, \alpha) - G_{\mathscr{X}}^2(x_i, \bar{f}_i, \alpha)\| = \|\frac{\sqrt{\hat{v}_i^1}}{\alpha}(x - x_{1,i}^+) - \frac{\sqrt{\hat{v}_i^2}}{\alpha}(x - x_{2,i}^+)\|$$

$$\leq \|\frac{\sqrt{\hat{v}_i^1}}{\alpha}(x - x_{1,i}^+) - \frac{\sqrt{\hat{v}_i^1}}{\alpha}(x - x_{2,i}^+)\| \overset{(i)}{\leq} \frac{\bar{v}}{\alpha}\|x_{2,i}^+ - x_{1,i}^+\| \overset{(ii)}{\leq} \frac{\bar{v}}{\underline{v}}\|m_i^1 - m_i^2\|,$$

where (i) follows from equation 11 and (ii) follows from equation 51. $\qquad \square$

**Lemma 20.** *Let $\beta_1, \beta_2 \in [0,1)$ satisfy $\eta = \frac{\beta_1}{\sqrt{\beta_2}} < 1$. Then, for the sequence $x_{i,t}$ generated by Algorithm 1, we have*

$$\frac{1}{n}\sum_{i=1}^n \|x_{i,t} - x_t^*\| \leq \frac{2\sqrt{n}}{(1-\eta)\sqrt{(1-\beta_2)}}\sum_{s=0}^{t-1}\alpha_s\sigma_2^{t-s-1}(W) := B_t.$$

*Proof.* In the proof of Lemma 15 we showed that

$$x_{i,t+1} - \bar{x}_{t+1} = \sum_{s=0}^t \sum_{j=1}^n ([W^{t-s}]_{ij} - \frac{1}{n})e_{i,s}.$$

Now, using the above equality, we have

$$\|x_{i,t+1} - \bar{x}_{t+1}\| = \sum_{s=0}^t \sum_{j=1}^n |[W^{t-s}]_{ij} - \frac{1}{n}|\|e_{i,s}\|. \tag{52}$$

Also, using Lemma 14, we get

$$\frac{m_{i,t}}{\sqrt{\hat{v}_{i,t}}} \leq \frac{1}{(1-\eta)\sqrt{(1-\beta_2)}}. \tag{53}$$

The above inequality together with the update rule of $x_{i,t+1}$, imply that

$$\|e_{i,t}\| = \|x_{i,t+1} - \sum_{j=1}^n [W]_{ij}x_{j,t}\| = \|\Pi_{\mathscr{X}}[\sum_{j=1}^n [W]_{ij}x_{j,t} - \alpha_t \frac{m_{i,t}}{\sqrt{\hat{v}_{i,t}}}] - \sum_{j=1}^n [W]_{ij}x_{j,t}\|$$

$$\leq \|\sum_{j=1}^n [W]_{ij}x_{j,t} - \alpha_t \frac{m_{i,t}}{\sqrt{\hat{v}_{i,t}}} - \sum_{j=1}^n [W]_{ij}x_{j,t}\|$$

$$= \alpha_t \|\frac{m_{i,t}}{\sqrt{\hat{v}_{i,t}}}\|$$

$$\leq \frac{\alpha_t}{(1-\eta)\sqrt{(1-\beta_2)}}, \tag{54}$$

where the first inequality follows from the nonexpansiveness property of the Euclidean projection[4]. Substituting equation 26 and equation 54 into equation 52, we get

$$\|x_{i,t} - \bar{x}_t\| \leq \frac{\sqrt{n}}{(1-\eta)\sqrt{(1-\beta_2)}}\sum_{s=0}^{t-1}\alpha_s\sigma_2^{t-s-1}(W).$$

Summing the above inequality over $i \in \mathscr{V}$, we conclude that

$$\sum_{i=1}^n \|x_{i,t} - x_t^*\| \leq \sum_{i=1}^n \|x_{i,t} - \bar{x}_t\| + \sum_{i=1}^n \|\bar{x}_t - x_t^*\| \leq \frac{2n\sqrt{n}}{(1-\eta)\sqrt{(1-\beta_2)}}\sum_{s=0}^{t-1}\alpha_s\sigma_2^{t-s-1}(W).$$

$\qquad \square$

---

[4]$\|\Pi_{\mathscr{X}}[x_1] - \Pi_{\mathscr{X}}[x_2]\| \leq \|x_1 - x_2\|, \ \forall x_1, x_2 \in \mathbb{R}^p.$

**Lemma 21.** *For the sequence $x_{i,t}$ generated by Algorithm 1, we have*

$$\langle \nabla \bar{f}_{i,t}, G_{\mathscr{X}}(x_{i,t}, \bar{f}_{i,t}, \alpha_t) \rangle \geq \frac{(2 - \beta_{1,t})}{2(1 - \beta_{1,t})} \|G_{\mathscr{X}}(x_{i,t}, \bar{f}_{i,t}, \alpha_t)\|^2 - \frac{\beta_{1,t} \hat{v}_{i,t}}{2(1 - \beta_{1,t})(1 - \eta)^2(1 - \beta_2)}.$$

*Proof.* A quick look at optimality condition of equation 6 verifies that

$$\langle \frac{m_{i,t}}{\sqrt{\hat{v}_{i,t}}} + \frac{1}{\alpha_t}(x_{i,t+1} - \sum_{j=1}^{n}[W]_{ij}x_{j,t}), z - x_{i,t+1} \rangle \geq 0, \qquad \forall z \in \mathscr{X}.$$

Substituting $z = x_{i,t}$ into the above inequality, we get

$$\langle \frac{m_{i,t}}{\sqrt{\hat{v}_{i,t}}}, x_{i,t} - x_{i,t+1} \rangle \geq \frac{1}{\alpha_t} \langle x_{i,t+1} - \sum_{j=1}^{n}[W]_{ij}x_{j,t}, x_{i,t+1} - x_{i,t} \rangle.$$

Now using equation 2, we have

$$\frac{\sqrt{\hat{v}_{i,t}}}{\alpha_t}\|x_{i,t} - x_{i,t+1}\|^2 \leq \langle m_{i,t}, x_{i,t} - x_{i,t+1} \rangle = \langle \beta_{1,t}m_{i,t-1} + (1 - \beta_{1,t})\nabla\bar{f}_{i,t}, x_{i,t} - x_{i,t+1} \rangle$$

$$= \frac{\beta_{1,t}\sqrt{\hat{v}_{i,t-1}}}{\alpha_t}\langle \frac{\alpha_t m_{i,t-1}}{\sqrt{\hat{v}_{i,t-1}}}, x_{i,t} - x_{i,t+1} \rangle + (1 - \beta_{1,t})\langle \nabla\bar{f}_{i,t}, x_{i,t} - x_{i,t+1} \rangle$$

$$\leq \frac{\beta_{1,t}\sqrt{\hat{v}_{i,t}}}{2\alpha_t}(\frac{\alpha_t^2}{(1 - \eta)^2(1 - \beta_2)} + \|x_{i,t} - x_{i,t+1}\|^2) + (1 - \beta_{1,t})\langle \nabla\bar{f}_{i,t}, x_{i,t} - x_{i,t+1} \rangle,$$

where the first equality follows from the update rule of $m_{i,t}$. The last inequality is valid since Cauchy-Schwarz inequality and inequality $\sqrt{\hat{v}_{i,t-1}}\beta_{1,t} \leq \sqrt{\hat{v}_{i,t}}\beta_1$. The claim then follows after using Definition 1. $\qquad\square$

### 6.1.3 Proof of Theorem 8

*Proof.* Let $\bar{f}_{i,t}(x_{i,t}) = \frac{1}{t}\sum_{s=1}^{t}f_{i,s}(x_{i,t})$. Using Assumption 2, Taylor's expansion and Definition 1, we have

$$\bar{f}_{i,t}(x_{i,t+1}) \leq \bar{f}_{i,t}(x_{i,t}) + \langle \nabla\bar{f}_{i,t}(x_{i,t}), x_{i,t+1} - x_{i,t} \rangle + \frac{\rho}{2}\|x_{i,t+1} - x_{i,t}\|^2$$

$$\leq \bar{f}_{i,t}(x_{i,t}) - \frac{\alpha_t}{\sqrt{\hat{v}_{i,t}}}\langle \nabla\bar{f}_{i,t}(x_{i,t}), G_{\mathscr{X}}(x_{i,t}, \bar{f}_{i,t}, \alpha_t) \rangle + \frac{\rho\alpha_t^2}{2\hat{v}_{i,t}}\|G_{\mathscr{X}}(x_{i,t}, \bar{f}_{i,t}, \alpha_t)\|^2.$$

The above inequality together with Lemma 21, equation 11 and $\beta_{1,t} \leq \beta_1$, imply that

$$\bar{f}_{i,t}(x_{i,t+1}) \leq \bar{f}_{i,t}(x_{i,t}) - \frac{\alpha_t(2 - \beta_1)}{2\bar{v}}\|G_{\mathscr{X}}(x_{i,t}, \bar{f}_{i,t}, \alpha_t)\|^2$$

$$+ \frac{\alpha_t\beta_{1,t}\bar{v}}{2(1 - \beta_{1,t})(1 - \eta)^2(1 - \beta_2)} + \frac{\rho\alpha_t^2}{2\underline{v}^2}\|G_{\mathscr{X}}(x_{i,t}, \bar{f}_{i,t}, \alpha_t)\|^2. \qquad (55)$$

Let $\Delta_{i,t} = \bar{f}_{i,t}(x_{i,t}) - \bar{f}_{i,t}(x_t^*)$ denotes the instantaneous loss at round $t$. We have

$$\Delta_{i,t+1} = \frac{t}{t+1}\underbrace{(\bar{f}_{i,t}(x_{i,t+1}) - \bar{f}_{i,t}(x_{t+1}^*))}_{(I)} + \frac{1}{t+1}(f_{i,t+1}(x_{i,t+1}) - f_{i,t+1}(x_{t+1}^*)). \qquad (56)$$

Note that $(I)$ can be bounded as follows:

$$\bar{f}_{i,t}(x_{i,t+1}) - \bar{f}_{i,t}(x_{t+1}^*) \leq \bar{f}_{i,t}(x_{i,t+1}) - \bar{f}_{i,t}(x_t^*) \leq \Delta_{i,t} - [\frac{(2 - \beta_1)\alpha_t}{2\bar{v}} - \frac{\rho\alpha_t^2}{2\underline{v}^2}]\|G_{\mathscr{X}}(x_{i,t}, \bar{f}_{i,t}, \alpha_t)\|^2$$

$$+ \frac{\alpha_t\beta_{1,t}\bar{v}}{2(1 - \beta_{1,t})(1 - \eta)^2(1 - \beta_2)}, \qquad (57)$$

where the first inequality is due to the optimality of $x_t^*$ and the second inequality follows from equation 55.

Now, combining equation 57 and equation 56 gives

$$\Delta_{i,t+1} \le \frac{t}{t+1}(\Delta_{i,t} - [\frac{(2-\beta_1)\alpha_t}{2\bar{v}} - \frac{\rho\alpha_t^2}{2\underline{v}^2}]\|G_{\mathscr{X}}(x_{i,t}, \bar{f}_{i,t}, \alpha_t)\|^2$$
$$+ \frac{\alpha_t\beta_{1,t}\bar{v}}{2(1-\beta_{1,t})(1-\eta)^2(1-\beta_2)}) + \frac{1}{t+1}(f_{i,t+1}(x_{i,t+1}) - f_{i,t+1}(x_{t+1}^*)).$$

By rearranging the above inequality, we have

$$\frac{t}{t+1}[\frac{(2-\beta_1)\alpha_t}{2\bar{v}} - \frac{\rho\alpha_t^2}{2\underline{v}^2}]\|G_{\mathscr{X}}(x_{i,t}, \bar{f}_{i,t}, \alpha_t)\|^2 \le \frac{t}{t+1}(\Delta_{i,t} + \frac{\alpha_t\beta_{1,t}\bar{v}}{2(1-\beta_{1,t})(1-\eta)^2(1-\beta_2)})$$
$$+ \frac{1}{t+1}(f_{i,t+1}(x_{i,t+1}) - f_{i,t+1}(x_{t+1}^*)) - \Delta_{i,t+1}. \quad (58)$$

Using the definition of $\Delta_{i,t+1}$, we get $\frac{1}{t+1}(f_{i,t+1}(x_{i,t+1}) - f_{i,t+1}(x_{t+1}^*)) - \Delta_{i,t+1} = -(\frac{t}{t+1})(\bar{f}_{i,t}(x_{i,t+1}) - \bar{f}_{i,t}(x_{t+1}^*))$. Hence, using equation 58 and simplifying terms, we obtain

$$[\frac{(2-\beta_1)\alpha_t}{2\bar{v}} - \frac{\rho\alpha_t^2}{2\underline{v}^2}]\|G_{\mathscr{X}}(x_{i,t}, \bar{f}_{i,t}, \alpha_t)\|^2$$
$$\le \Delta_{i,t} - (\bar{f}_{i,t}(x_{i,t+1}) - \bar{f}_{i,t}(x_{t+1}^*)) + \frac{\alpha_t\beta_{1,t}\bar{v}}{2(1-\beta_{1,t})(1-\eta)^2(1-\beta_2)}. \quad (59)$$

Note that:

$$\frac{1}{n}\sum_{i=1}^{n}\sum_{t=1}^{T}\left(\Delta_{i,t} - (\bar{f}_{i,t}(x_{i,t+1}) - \bar{f}_{i,t}(x_{t+1}^*))\right)$$
$$= \frac{1}{n}\sum_{i=1}^{n}\sum_{t=1}^{T}\left((\bar{f}_{i,t}(x_{i,t}) - \bar{f}_{i,t}(x_{i,t+1})) - (\bar{f}_{i,t}(x_t^*) - \bar{f}_{i,t}(x_{t+1}^*))\right)$$
$$= \frac{1}{n}\sum_{i=1}^{n}[-\bar{f}_{i,T}(x_{i,T+1}) + \bar{f}_{i,1}(x_{i,1}) - \bar{f}_{i,1}(x_1^*) + \bar{f}_{i,T}(x_{T+1}^*)]$$
$$+ \frac{1}{n}\sum_{i=1}^{n}\sum_{t=2}^{T}t^{-1}\left(f_{i,t}(x_{i,t}) - \bar{f}_{i,t-1}(x_{i,t}) - (f_{i,t}(x_t^*) - \bar{f}_{i,t-1}(x_t^*))\right) \quad (60)$$
$$\le \frac{1}{n}\sum_{i=1}^{n}L\left(\|x_{i,T+1} - x_{T+1}^*\| + \|x_{i,1} - x_1^*\| + \sum_{t=2}^{T}2t^{-1}\|x_{i,t} - x_t^*\|\right) \quad (61)$$
$$\le 2L\max_{t\in\{2,...,T\}}B_t(1 + \sum_{t=2}^{T}t^{-1}) \le 2L\max_{t\in\{2,...,T\}}B_t(2 + \log T), \quad (62)$$

where the first equality uses

$$\bar{f}_{i,t}(x_{i,t}) - \bar{f}_{i,t-1}(x_{i,t}) = t^{-1}(f_{i,t}(x_{i,t}) - \bar{f}_{i,t-1}(x_{i,t})).$$

The first inequality follows from Assumption 2. The second inequality follows from Lemma 20 and equation 19.

Summing equation 59 over $i \in \mathscr{V}$, $t \in \{1,\ldots,T\}$ and using equation 62, we have

$$\frac{1}{n}\sum_{i=1}^{n}\min_{t\in\{1,...,T\}}\|G_{\mathscr{X}}(x_{i,t}, \bar{f}_{i,t}, \alpha_t)\|^2 \sum_{t=1}^{T}[\frac{(2-\beta_1)\alpha_t}{2\bar{v}} - \frac{\rho\alpha_t^2}{2\underline{v}^2}]$$
$$\le \frac{1}{n}\sum_{i=1}^{n}\sum_{t=1}^{T}[\frac{(2-\beta_1)\alpha_t}{2\bar{v}} - \frac{\rho\alpha_t^2}{2\underline{v}^2}]\|G_{\mathscr{X}}(x_{i,t}, \bar{f}_{i,t}, \alpha_t)\|^2$$
$$\le (2 + \log T)2L\max_{t\in\{2,...,T\}}\frac{2\sqrt{n}}{(1-\eta)\sqrt{(1-\beta_2)}}\sum_{s=0}^{t-1}\alpha_s\sigma_2^{t-s-1}(W)$$
$$+ \sum_{t=1}^{T}\frac{\alpha_t\beta_{1,t}\bar{v}}{2(1-\beta_{1,t})(1-\eta)^2(1-\beta_2)}. \quad (63)$$

Note that $\sum_{t=1}^{T} \left[ \frac{(2-\beta_1)\alpha_t}{2\bar{v}} - \frac{\rho\alpha_t^2}{2\underline{v}^2} \right] > 0$. Therefore, dividing both sides of the equation 63 by $\sum_{t=1}^{T} \left[ \frac{(2-\beta_1)\alpha_t}{2\bar{v}} - \frac{\rho\alpha_t^2}{2\underline{v}^2} \right]$, we obtain equation 14. $\qquad\square$

### 6.1.4   PROOF OF COROLLARY 9

*Proof.* With the constant step-sizes $\alpha_t = \frac{(2-\beta_1)\underline{v}^2}{2\rho\bar{v}}$ for all $t \in \{1, \ldots, T\}$, we have

$$\vartheta_t = \sum_{t=1}^{T} \left[ \frac{(2-\beta_1)\alpha_t}{2\bar{v}} - \frac{\rho\alpha_t^2}{2\underline{v}^2} \right] = \frac{T(2-\beta_1)^2\underline{v}^2}{8\rho\bar{v}^2}.$$

Therefore, using the above equality and equation 12 together with $\alpha_t = \frac{(2-\beta_1)\underline{v}^2}{2\rho\bar{v}}$ for all $t \in \{1, \ldots, T\}$, we obtain

$$
\begin{aligned}
\frac{1}{\vartheta_t} \sum_{t=1}^{T} \frac{\alpha_t \beta_{1,t} \bar{v}}{2(1-\beta_{1,t})(1-\eta)^2(1-\beta_2)} &= \frac{4\bar{v}}{T(2-\beta_1)} \sum_{t=1}^{T} \frac{\beta_{1,t} \bar{v}}{2(1-\beta_{1,t})(1-\eta)^2(1-\beta_2)} \\
&\leq \frac{4\bar{v}}{T(2-\beta_1)} \sum_{t=1}^{T} \frac{\lambda^{t-1} \bar{v}}{2(1-\beta_1)(1-\eta)^2(1-\beta_2)} \\
&\leq \frac{2\bar{v}^2}{T(2-\beta_1)(1-\beta_1)(1-\eta)^2(1-\beta_2)(1-\lambda)}, \qquad (64)
\end{aligned}
$$

and

$$
\begin{aligned}
\frac{1}{\vartheta_t}(2+\log T)2L \max_{t \in \{2,\ldots,T\}} \frac{2\sqrt{n}}{(1-\eta)\sqrt{(1-\beta_2)}} \sum_{s=0}^{t-1} \alpha_s \sigma_2^{t-s-1}(W) & \\
\leq \frac{16\bar{v}(2+\log T)L\sqrt{n}}{T(2-\beta_1)(1-\eta)\sqrt{(1-\beta_2)}} \max_{t \in \{2,\ldots,T\}} \sum_{s=0}^{t-1} \sigma_2^{t-s-1}(W) & \\
\leq \frac{16\bar{v}(2+\log T)L\sqrt{n}}{T(2-\beta_1)(1-\eta)\sqrt{(1-\beta_2)}(1-\sigma_2(W))}. & \qquad (65)
\end{aligned}
$$

Combine the results in equation 64 and equation 65, we have equation 13. $\qquad\square$

### 6.1.5   PROOF OF THEOREM 10

*Proof.* Let

$$\delta_{i,t} = \nabla \bar{f}_{i,t}(x_{i,t}) - \nabla \tilde{f}_{i,t}(x_{i,t}), \qquad \forall t \geq 1,$$

where $\nabla \bar{f}_{i,t}(x_{i,t}) = \frac{1}{t} \sum_{s=1}^{t} \nabla f_{i,s}(x_{i,t})$ denotes the minibatch stochastic gradient on node $i$.

Since $f_{i,t}$ is $\rho$-smooth, it follows that $\bar{f}_{i,t}$ is also $\rho$-smooth. Hence for every $i \in \mathcal{V}$ and $t \in \{1, \ldots, T\}$, we have

$$
\begin{aligned}
\bar{f}_{i,t}(x_{i,t+1}) &\leq \bar{f}_{i,t}(x_{i,t}) + \langle \nabla \bar{f}_{i,t}(x_{i,t}), x_{i,t+1} - x_{i,t} \rangle + \frac{\rho}{2} \|x_{i,t+1} - x_{i,t}\|^2 \\
&\leq \bar{f}_{i,t}(x_{i,t}) - \frac{\alpha_t}{\sqrt{\hat{v}_{i,t}}} \langle \nabla \bar{f}_{i,t}(x_{i,t}), \mathbf{G}_{\mathcal{X}}(x_{i,t}, \bar{f}_{i,t}, \alpha_t) \rangle + \frac{\rho\alpha_t^2}{2\hat{v}_{i,t}} \|\mathbf{G}_{\mathcal{X}}(x_{i,t}, \bar{f}_{i,t}, \alpha_t)\|^2 \\
&= \bar{f}_{i,t}(x_{i,t}) - \frac{\alpha_t}{\sqrt{\hat{v}_{i,t}}} \langle \nabla \bar{f}_{i,t}(x_{i,t}), \mathbf{G}_{\mathcal{X}}(x_{i,t}, \bar{f}_{i,t}, \alpha_t) \rangle + \frac{\rho\alpha_t^2}{2\hat{v}_{i,t}} \|\mathbf{G}_{\mathcal{X}}(x_{i,t}, \bar{f}_{i,t}, \alpha_t)\|^2 \\
&\quad + \frac{\alpha_t}{\sqrt{\hat{v}_{i,t}}} \langle \delta_{i,t}, \mathbf{G}_{\mathcal{X}}(x_{i,t}, \bar{f}_{i,t}, \alpha_t) \rangle,
\end{aligned}
$$

where $\mathbf{G}_{\mathcal{X}}(x_{i,t}, \bar{f}_{i,t}, \alpha_t)$ denotes the projected stochastic gradient on node $i$.

The above inequality together with Lemma 21, equation 11 and $\beta_{1,t} \leq \beta_1$, imply that

$$\bar{f}_{i,t}(x_{i,t+1}) \leq \bar{f}_{i,t}(x_{i,t}) - \frac{\alpha_t(2-\beta_1)}{2\bar{v}}\|\mathbf{G}_{\mathscr{X}}(x_{i,t},\bar{f}_{i,t},\alpha_t)\|^2 + \frac{\alpha_t\beta_{1,t}\bar{v}}{2(1-\beta_{1,t})(1-\eta)^2(1-\beta_2)}$$

$$+ \frac{\rho\alpha_t^2}{2\underline{v}^2}\|G_{\mathscr{X}}(x_{i,t},\bar{f}_{i,t},\alpha_t)\|^2 + \frac{\alpha_t}{\sqrt{\hat{v}_{i,t}}}\langle\delta_{i,t},G_{\mathscr{X}}(x_{i,t},\bar{f}_{i,t},\alpha_t)\rangle$$

$$+ \frac{\alpha_t}{\sqrt{\hat{v}_{i,t}}}\langle\delta_{i,t},\mathbf{G}_{\mathscr{X}}(x_{i,t},\bar{f}_{i,t},\alpha_t) - G_{\mathscr{X}}(x_{i,t},\bar{f}_{i,t},\alpha_t)\rangle$$

$$\leq \bar{f}_{i,t}(x_{i,t}) - \frac{\alpha_t(2-\beta_1)}{2\bar{v}}\|\mathbf{G}_{\mathscr{X}}(x_{i,t},\bar{f}_{i,t},\alpha_t)\|^2 + \frac{\alpha_t\beta_{1,t}\bar{v}}{2(1-\beta_{1,t})(1-\eta)^2(1-\beta_2)}$$

$$+ \frac{\rho\alpha_t^2}{2\underline{v}^2}\|G_{\mathscr{X}}(x_{i,t},\bar{f}_{i,t},\alpha_t)\|^2 + \frac{\alpha_t}{\sqrt{\hat{v}_{i,t}}}\langle\delta_{i,t},G_{\mathscr{X}}(x_{i,t},\bar{f}_{i,t},\alpha_t)\rangle$$

$$+ \frac{\alpha_t}{\sqrt{\hat{v}_{i,t}}}\|\delta_{i,t}\|\|\mathbf{G}_{\mathscr{X}}(x_{i,t},\bar{f}_{i,t},\alpha_t) - G_{\mathscr{X}}(x_{i,t},\bar{f}_{i,t},\alpha_t)\|$$

$$\leq \bar{f}_{i,t}(x_{i,t}) - \frac{\alpha_t(2-\beta_1)}{2\bar{v}}\|\mathbf{G}_{\mathscr{X}}(x_{i,t},\bar{f}_{i,t},\alpha_t)\|^2 + \frac{\alpha_t\beta_{1,t}\bar{v}}{2(1-\beta_{1,t})(1-\eta)^2(1-\beta_2)}$$

$$+ \frac{\rho\alpha_t^2}{2\underline{v}^2}\|G_{\mathscr{X}}(x_{i,t},\bar{f}_{i,t},\alpha_t)\|^2 + \frac{\alpha_t}{\sqrt{\hat{v}_{i,t}}}\langle\delta_{i,t},G_{\mathscr{X}}(x_{i,t},\bar{f}_{i,t},\alpha_t)\rangle$$

$$+ \frac{\alpha_t}{\sqrt{\hat{v}_{i,t}}}\|\delta_{i,t}\|\frac{\bar{v}}{\underline{v}}\sum_{r=1}^{t}(1-\beta_{1,r})\beta_1^{t-r}\|\delta_{i,r}\|, \tag{66}$$

where the second inequality follows from the Cauchy-Schwartz inequality. The last inequality follows from the fact that $\beta_{1,t} = \beta_1\lambda^{t-1}, \lambda \in (0,1)$ and Lemma 19 if we set $G_{\mathscr{X}}^1(x_i,\bar{f}_i,\alpha) = \mathbf{G}_{\mathscr{X}}(x_{i,t},\bar{f}_{i,t},\alpha_t)$, and $G_{\mathscr{X}}^2(x_i,\bar{f}_i,\alpha) = G_{\mathscr{X}}(x_{i,t},\bar{f}_{i,t},\alpha_t)$.

Let $\Delta_{i,t} = \bar{f}_{i,t}(x_{i,t}) - \bar{f}_{i,t}(x_t^*)$ denotes the instantaneous stochastic loss at time $t$. We have

$$\Delta_{i,t+1} = \frac{t}{t+1}\underbrace{(\bar{f}_{i,t}(x_{i,t+1}) - \bar{f}_{i,t}(x_{t+1}^*))}_{(I)} + \frac{1}{t+1}(f_{i,t+1}(x_{i,t+1}) - f_{i,t+1}(x_{t+1}^*)). \tag{67}$$

Observe that $(I)$ can be bounded as follows:

$$\bar{f}_{i,t}(x_{i,t+1}) - \bar{f}_{i,t}(x_{t+1}^*) \leq \bar{f}_{i,t}(x_{i,t+1}) - \bar{f}_{i,t}(x_t^*) \leq \Delta_{i,t} - [\frac{(2-\beta_1)\alpha_t}{2\bar{v}} - \frac{\rho\alpha_t^2}{2\underline{v}^2}]\|G_{\mathscr{X}}(x_{i,t},\bar{f}_{i,t},\alpha_t)\|^2$$

$$+ \frac{\alpha_t\beta_{1,t}\bar{v}}{2(1-\beta_{1,t})(1-\eta)^2(1-\beta_2)} + \frac{\alpha_t}{\sqrt{\hat{v}_{i,t}}}\langle\delta_{i,t},G_{\mathscr{X}}(x_{i,t},\bar{f}_{i,t},\alpha_t)\rangle + \frac{\bar{v}\alpha_t}{\underline{v}^2}\sum_{r=1}^{t}\beta_1^{t-r}\|\delta_{i,r}\|\|\delta_{i,t}\|, \tag{68}$$

where the first inequality is due to $x_{t+1}^* \in \mathscr{X}$ and the optimality of $x_t^*$ and the second inequality is due to equation 66 and the fact that $0 \leq \beta_{1,r} < 1$. Thus, using equation 67 and equation 68, we get

$$\Delta_{i,t+1} \leq \frac{t}{t+1}\Big(\Delta_{i,t} - [\frac{(2-\beta_1)\alpha_t}{2\bar{v}} - \frac{\rho\alpha_t^2}{2\underline{v}^2}]\|G_{\mathscr{X}}(x_{i,t},\bar{f}_{i,t},\alpha_t)\|^2 + \frac{\alpha_t\beta_{1,t}\bar{v}}{2(1-\beta_{1,t})(1-\eta)^2(1-\beta_2)}$$

$$+ \frac{\alpha_t}{\sqrt{\hat{v}_{i,t}}}\langle\delta_{i,t},G_{\mathscr{X}}(x_{i,t},\bar{f}_{i,t},\alpha_t)\rangle + \frac{\bar{v}\alpha_t}{\underline{v}^2}\sum_{r=1}^{t}\beta_1^{t-r}\|\delta_{i,r}\|\|\delta_{i,t}\|\Big) + \frac{1}{t+1}(f_{i,t+1}(x_{i,t+1}) - f_{i,t+1}(x_{t+1}^*)).$$

By rearranging the above inequality, we have

$$\frac{t}{t+1}[\frac{(2-\beta_1)\alpha_t}{2\bar{v}} - \frac{\rho\alpha_t^2}{2\underline{v}^2}]\|G_{\mathscr{X}}(x_{i,t},\bar{f}_{i,t},\alpha_t)\|^2 \leq \frac{t}{t+1}\Big(\Delta_{i,t} + \frac{\alpha_t\beta_{1,t}\bar{v}}{2(1-\beta_{1,t})(1-\eta)^2(1-\beta_2)}$$

$$+ \frac{\alpha_t}{\sqrt{\hat{v}_{i,t}}}\langle\delta_{i,t},G_{\mathscr{X}}(x_{i,t},\bar{f}_{i,t},\alpha_t)\rangle + \frac{\bar{v}\alpha_t}{\underline{v}^2}\sum_{r=1}^{t}\beta_1^{t-r}\|\delta_{i,r}\|\|\delta_{i,t}\|\Big)$$

$$+ \frac{1}{t+1}(f_{i,t+1}(x_{i,t+1}) - f_{i,t+1}(x_{t+1}^*)) - \Delta_{i,t+1}. \tag{69}$$

Now, using definition of $\Delta_{i,t+1}$, $\frac{1}{t+1}(f_{i,t+1}(x_{i,t+1}) - f_{i,t+1}(x^*_{t+1})) - \Delta_{i,t+1} = -(\frac{t}{t+1})(\bar{f}_{i,t}(x_{i,t+1}) - \bar{f}_{i,t}(x^*_{t+1}))$, together with equation 69, we obtain

$$[\frac{(2-\beta_1)\alpha_t}{2\bar{v}} - \frac{\rho\alpha_t^2}{2\underline{v}^2}]\|G_{\mathscr{X}}(x_{i,t}, \bar{f}_{i,t}, \alpha_t)\|^2 \leq \Delta_{i,t} - (\bar{f}_{i,t}(x_{i,t+1}) - \bar{f}_{i,t}(x^*_{t+1}))$$

$$+ \frac{\alpha_t\beta_{1,t}\bar{v}}{2(1-\beta_{1,t})(1-\eta)^2(1-\beta_2)} + \frac{\alpha_t}{\sqrt{\hat{v}_{i,t}}}\langle\delta_{i,t}, G_{\mathscr{X}}(x_{i,t}, \bar{f}_{i,t}, \alpha_t)\rangle + \frac{\bar{v}\alpha_t}{\underline{v}^2}\sum_{r=1}^{t}\beta_1^{t-r}\|\delta_{i,r}\|\|\delta_{i,t}\|. \quad (70)$$

Note that from Assumption 3, we have $\mathbb{E}\left[\langle\delta_{i,t}, G_{\mathscr{X}}(x_{i,t}, \bar{f}_{i,t}, \alpha_t)\rangle\right] = 0$ and

$$\mathbb{E}\left[\|\nabla f_{i,t}(x_{i,t}) - \nabla f_{i,t}(x_{i,t})\|^2\right] = \mathbb{E}\left[\|\nabla f_{i,t}(x_{i,t})\|^2\right] - \|\mathbb{E}[\nabla f_{i,t}(x_{i,t})]\|^2 \leq \mathbb{E}\left[\|\nabla f_{i,t}(x_{i,t})\|^2\right] \leq \xi^2,$$

which implies that

$$\mathbb{E}\left[\|\delta_{i,t}\|^2\right] = \mathbb{E}\left[\|\nabla\bar{f}_{i,t}(x_{i,t}) - \nabla\bar{f}_{i,t}(x_{i,t})\|^2\right]$$

$$= \mathbb{E}\left[\|\frac{1}{t}\sum_{s=1}^{t}(\nabla f_{i,s}(x_{i,t}) - \nabla f_{i,s}(x_{i,t}))\|^2\right]$$

$$\leq \frac{1}{t}\sum_{s=1}^{t}\mathbb{E}\left[\|\nabla f_{i,s}(x_{i,t}) - \nabla f_{i,s}(x_{i,t})\|^2\right] \leq \xi^2.$$

The above inequality together with Cauchy-Schwarz for expextations, imply that

$$\mathbb{E}[\|\delta_{i,t}\|\|\delta_{i,r}\|] \leq \sqrt{\mathbb{E}[\|\delta_{i,r}\|^2]}\sqrt{\mathbb{E}[\|\delta_{i,t}\|^2]} \leq \xi^2. \quad (71)$$

Now, using equation 71 and equation 62, summing equation 70 over $i \in \mathscr{V}$ and $t \in \{1,\ldots,T\}$, and taking expectation, we obtain

$$\frac{1}{n}\sum_{i=1}^{n}\min_{t\in\{1,\ldots,T\}}\mathbb{E}\left[\|\mathbf{G}_{\mathscr{X}}(x_{i,t}, \bar{f}_{i,t}, \alpha_t)\|^2\right]\sum_{t=1}^{T}[\frac{(2-\beta_1)\alpha_t}{2\bar{v}} - \frac{\rho\alpha_t^2}{2\underline{v}^2}]$$

$$\leq \sum_{t=1}^{T}[\frac{(2-\beta_1)\alpha_t}{2\bar{v}} - \frac{\rho\alpha_t^2}{2\underline{v}^2}]\frac{1}{n}\sum_{i=1}^{n}\mathbb{E}\left[\|\mathbf{G}_{\mathscr{X}}(x_{i,t}, \bar{f}_{i,t}, \alpha_t)\|^2\right]$$

$$\leq (2+\log T)2L\max_{t\in\{2,\ldots,T\}}\frac{2\sqrt{n}}{(1-\eta)\sqrt{(1-\beta_2)}}\sum_{s=0}^{t-1}\alpha_s\sigma_2^{t-s-1}(W)$$

$$+ \sum_{t=1}^{T}\frac{\alpha_t\beta_{1,t}\bar{v}}{2(1-\beta_{1,t})(1-\eta)^2(1-\beta_2)} + \frac{\bar{v}\xi^2}{\underline{v}^2}\sum_{t=1}^{T}\alpha_t\sum_{r=1}^{t}\beta_1^{t-r}. \quad (72)$$

Note that $\sum_{t=1}^{T}[\frac{(2-3\beta_1)\alpha_t}{2\bar{v}} - \frac{\rho\alpha_t^2}{2\underline{v}^2}] > 0$, and

$$\sum_{t=1}^{T}\alpha_t\sum_{r=1}^{t}\beta_1^{t-r} = \sum_{t=1}^{T}\alpha_t\sum_{r=t}^{T}\beta_1^{r-t} \leq \sum_{t=1}^{T}\frac{\alpha_t}{(1-\beta_1)}. \quad (73)$$

Therefore, dividing both sides of the equation 75 by $\sum_{t=1}^{T}[\frac{(2-3\beta_1)\alpha_t}{2\bar{v}} - \frac{\rho\alpha_t^2}{2\underline{v}^2}]$, using equation 73, we complete the proof. $\qquad\square$

### 6.1.6 Proof of Corollary 11

*Proof.* Let $\vartheta_t = \sum_{t=1}^{T}[\frac{(2-\beta_1)\alpha_t}{2\bar{v}} - \frac{\rho\alpha_t^2}{2\underline{v}^2}]$. We claim that if

$$T \geq \max\{\frac{4\rho^2\bar{v}^2}{n\underline{v}^4(2-\beta_1)^2}, \frac{4\bar{v}^2n}{(2-\beta_1)^2}\}, \quad (74)$$

then $\vartheta_t \geq 1/2$. One can easily seen that $T$ must satisfy $\frac{(2-\beta_1)\alpha_t}{2\bar{v}} - \frac{\rho\alpha_t^2}{2\underline{v}^2} \geq \frac{1}{2T}$ which results in

$$-T\rho\alpha_t^2\bar{v} + T\underline{v}^2(2-\beta_1)\alpha_t - \bar{v}\underline{v}^2 \geq 0.$$

By rearranging the above inequality, we obtain

$$1 \geq \frac{\rho \alpha_t \bar{\upsilon}}{\underline{\upsilon}^2(2-\beta_1)} + \frac{\bar{\upsilon}}{T(2-\beta_1)\alpha_t} := A_1 + A_2.$$

Now, if $A_1, A_2 \leq \frac{1}{2}$, then $\alpha_t \leq \frac{\underline{\upsilon}^2(2-\beta_1)}{2\rho \bar{\upsilon}}$, and $\alpha_t \geq \frac{2\bar{\upsilon}}{T(2-\beta_1)}$, respectively. These bounds together with our assumption on $\alpha_t$ give equation 74.

Now, let equation 74 holds. Using equation 70, and since $\vartheta_t \geq 1/2$, we have

$$\frac{1}{Tn} \sum_{i=1}^{n} \min_{t \in \{1,\dots,T\}} \mathbb{E}\left[\|\mathbf{G}_{\mathscr{X}}(x_{i,t}, \bar{f}_{i,t}, \alpha_t)\|^2\right]$$

$$\leq \frac{2}{Tn} \sum_{i=1}^{n} \min_{t \in \{1,\dots,T\}} \mathbb{E}\left[\|\mathbf{G}_{\mathscr{X}}(x_{i,t}, \bar{f}_{i,t}, \alpha_t)\|^2\right] \sum_{t=1}^{T} \left[\frac{(2-\beta_1)\alpha_t}{2\bar{\upsilon}} - \frac{\rho \alpha_t^2}{2\underline{\upsilon}^2}\right]$$

$$\leq \frac{2}{T} \sum_{t=1}^{T} \left[\frac{(2-\beta_1)\alpha_t}{2\bar{\upsilon}} - \frac{\rho \alpha_t^2}{2\underline{\upsilon}^2}\right] \frac{1}{n} \sum_{i=1}^{n} \mathbb{E}\left[\|\mathbf{G}_{\mathscr{X}}(x_{i,t}, \bar{f}_{i,t}, \alpha_t)\|^2\right]$$

$$\leq \frac{2}{Tn} \sum_{i=1}^{n} \left(f_{i,1}(x_{i,1}) - f_{i,1}(x_1^*)\right) + L\left(\|x_{i,T+1} - x_{T+1}^*\| + \sum_{t=2}^{T} 2t^{-1}\|x_{i,t} - x_t^*\|\right)$$

$$+ \frac{2}{T} \sum_{t=1}^{T} \frac{\alpha_t \beta_{1,t} \bar{\upsilon}}{2(1-\beta_{1,t})(1-\eta)^2(1-\beta_2)} + \frac{2\bar{\upsilon}\xi^2}{\underline{\upsilon}^2(1-\beta_1)T} \sum_{t=1}^{T} \alpha_t, \tag{75}$$

where the last inequality follows from equation 60, equation 61 and equation 71.

We proceed to bound RHS of equation 75. By substituting $\alpha_t = \frac{\alpha}{\sqrt{nT}}$ and $\beta_{1,t} = \beta_1 \lambda^{t-1}, \lambda \in (0,1)$ into the last two terms of equation 75, we have

$$\frac{2\bar{\upsilon}\xi^2}{\underline{\upsilon}^2(1-\beta_1)T} \sum_{t=1}^{T} \alpha_t = \frac{2\alpha \bar{\upsilon}\xi^2}{\underline{\upsilon}^2(1-\beta_1)\sqrt{nT}}, \tag{76}$$

and

$$\frac{1}{T} \sum_{t=1}^{T} \frac{\alpha_t \beta_{1,t} \bar{\upsilon}}{(1-\beta_{1,t})(1-\eta)^2(1-\beta_2)} = \frac{\alpha \bar{\upsilon}}{T\sqrt{nT}(1-\eta)^2(1-\beta_2)} \sum_{t=1}^{T} \frac{\beta_{1,t}}{(1-\beta_{1,t})}$$

$$= \frac{\alpha \bar{\upsilon}}{T\sqrt{nT}(1-\eta)^2(1-\beta_2)(1-\beta_1)(1-\lambda)}. \tag{77}$$

Further, using equation 75 and Lemma 20, we get

$$\frac{L}{Tn} \sum_{i=1}^{n} \|x_{i,T+1} - x_{T+1}^*\| \leq \frac{2\sqrt{n}\alpha L}{T(1-\eta)\sqrt{(1-\beta_2)}\sqrt{nT}} \sum_{s=0}^{T} \sigma_2^{T-s}(W)$$

$$\leq \frac{2\sqrt{n}\alpha L}{T(1-\eta)\sqrt{(1-\beta_2)}\sqrt{nT}(1-\sigma_2(W))}. \tag{78}$$

Similarly, using equation 75 and Lemma 20, we have

$$\frac{L}{Tn} \sum_{i=1}^{n} \sum_{t=2}^{T} t^{-1}\|x_{i,t} - x_t^*\| \leq \frac{2\sqrt{n}\alpha L}{\sqrt{nT}(1-\eta)\sqrt{(1-\beta_2)}T} \sum_{t=2}^{T} t^{-1} \sum_{s=0}^{t-1} \sigma_2^{t-s-1}(W)$$

$$\leq \frac{2\sqrt{n}\alpha L}{\sqrt{nT}(1-\eta)\sqrt{(1-\beta_2)}T} \sqrt{\sum_{t=2}^{T} t^{-2}} \sqrt{\sum_{t=2}^{T} \left(\sum_{s=0}^{t-1} \sigma_2^{t-s-1}(W)\right)^2}$$

$$\leq \frac{2\sqrt{n}\alpha L}{\sqrt{nT}(1-\eta)\sqrt{(1-\beta_2)}T(1-\sigma_2(W))}, \tag{79}$$

where the second inequality is due to Cauchy-Schwarz inequality and the last inequality follows because $\sum_{t=2}^{T} \frac{1}{t^2} \leq 1$.

Using equation 16a, equation 77, equation 78 and equation 79 are bounded by equation 76. This completes the proof. $\qquad\square$

## 6.2 SENSITIVITY OF DADAM TO ITS PARAMETERS

Next, we examine the sensitivity of DADAM on the parameters related to the network connection and update of the moment estimate. As it is clear from Figure 3 the convergence of training accuracy value happens faster for sparser networks (higher $\sigma_2(W)$). This is similar to the trend observed for FedAvg algorithm while reducing parameter $C$ which makes the agent interaction matrix sparser. This is also expected as discussed in Theorems 4 and 5. Note that with the availability of a central parameter server (as in FedAvg algorithm), sparser topology may be useful for a faster convergence, however, topology density of graph is important for a distributed learning scheme with decentralized computation on a network.

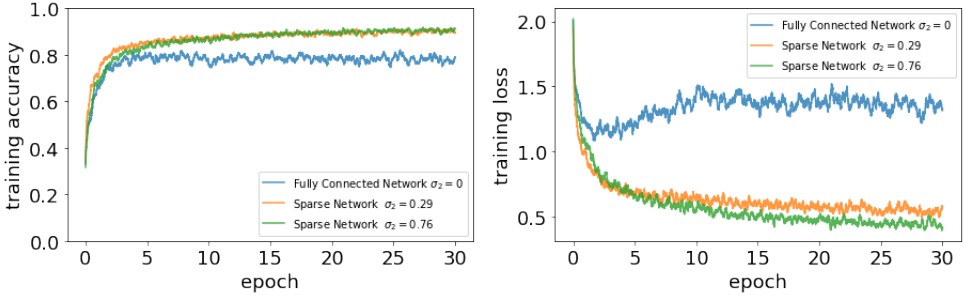

Figure 3: Performance of the DADAM algorithm with varying network topology: training loss and accuracy over 30 epochs based on the MNIST digit recognition library.

We also empirically evaluate the effect of the $\beta_3$ in Algorithm 1. We consider a range of hyper-parameter choices, i.e. $\beta_3 \in \{0, 0.9, 0.99\}$. From Figure 4 it can be easily seen that DADAM per-

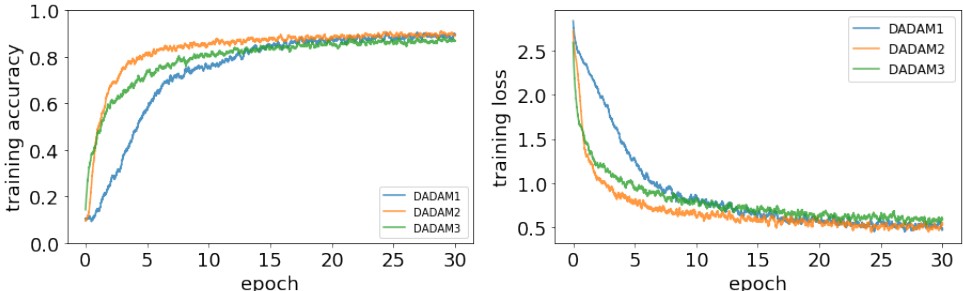

Figure 4: Performance of the DADAM algorithm with varying decay rate $\beta_3$. DADAM1 ($\beta_3 = 0$), DADAM2 ($\beta_3 = 0.9$), and DADAM3 ($\beta_3 = 0.99$) for training loss and accuracy over 30 epochs based on the MNIST digit recognition library.

forms equal or better than AMSGrad ($\beta_3 = 0$), regardless of the hyper-parameter setting for $\beta_1$ and $\beta_2$.

