# OpenReview forum: "DADAM: A consensus-based distributed adaptive gradient method for online optimization"
_ICLR.cc/2019/Conference_

### Official Review · AnonReviewer2 · 2018-10-27
**A consensus-based distributed adaptive gradient method for online optimization**

**Rating:** 6
**Confidence:** 4

**Review:**

Title: DADAM: A consensus-based distributed adaptive gradient method for online optimization

Summary:

The paper presented DADAM, a new consensus-based distributed adaptive moment estimation method, for online optimization. The author(s) also provide the convergence analysis and dynamic regret bound. The experiments show good performance of DADAM comparing to other methods.

Comments:

1) The theoretical results are nice and indeed non-trivial. However, could you please explain the implication to equation (7a)? Does it have absolute value on the LHS?

2) Can you explain more clearly about the section 3.2.1? It is not clear to me why DADAM outperform ADAM here.

3) Did you perform algorithms on many runs and take the average? Also, did you tune the learning rate for all other algorithms to be the best performance? I am not sure how you choose the parameter \alpha here. What if \alpha changes and do not base on that in Yuan et al. 2016?

4) The deep learning experiments are quite simple. In order to validate the performance of the algorithm, it needs to be run on more datasets and networks architectures. MNIST and CIFAR-10 and these simple network architectures are quite standard. I would suggest to provide more if the author(s) have time.

In general, I like this paper. I would love to have discussions with the author(s) during the rebuttal period.

---

> ### Author Response · Authors · 2018-11-17
> **Response to Reviewer~2**
>
>
>
> We thank the reviewer for very helpful and constructive feedback. A detailed point-by-point response to the reviewer's comments follows.
>
>
> 1-1 [Comment]
>
> -- Could you please explain the implication to equation (7a)? Does it have absolute value on the LHS?
>
> 1-1 [Response]
>
> -- Modified and Fixed.
>
>
>
> 1-2 [Comment]
>
> -- Can you explain more clearly about the section 3.2.1? It is not clear to me why DADAM outperform ADAM here.
>
> 1-2 [Response]
>
> -- In this section, we address the question of whether DADAM be faster than ADAM (which is a centralized adaptive algorithm)? We provide the analysis for the convergence rate of the stochastic DADAM in the non-convex setting and show that the convergence rate of DADAM w.r.t time steps is similar to the mini-batch SGD, mini-batch ADAM, centralized parallel ADAM and parallel stochastic gradient descent, but DADAM avoids the communication traffic jam due to its locally distributed nature.
>
> - In the context of stochastic nonconvex optimization, we say a gradient-based method gives an $\epsilon$-approximation solution if $$T^{-1}\e{{\bf Reg}^N_T} \leq \epsilon$$ where ${\bf Reg}^N_T$ is defined in Section~2 (please see Definition~2). Now, assume that $T$ is sufficiently large, it satisfies (16a) and (16b), the $\frac{1}{T}$ term will be dominated by the $\frac{1}{\sqrt{nT}}$ term which leads to a $\frac{1}{\sqrt{nT}}$ convergence rate. More specifically, it shows that the computation complexity of DADAM to achieve $\epsilon$-approximation solution is  $O(1/\epsilon^2)$. It is worth to mention that the computational complexity per iteration of DADAM is $O(n)$ since the computation of a single stochastic gradient counts 1. Further, since the total number of nodes does not affect the complexity, each node exhibits complexity of $O\big(1/(n\epsilon^2)\big)$.
>
> In summary, a linear speed up can be achieved by DADAM w.r.t computational complexity if $T$ is sufficiently large.
>
>
> Refer to Subsection~3.2.1 on page 8.
>
> 1-3 [Comment]
>
> Did you perform algorithms on many runs and take the average? Also, did you tune the learning rate for all other algorithms to be the best performance? I am not sure how you choose the parameter $\alpha$ here. What if $\alpha$ changes and do not base on that in Yuan et al. 2016?
>
> 1-3 [Response]
>
> -- The experiment is repeated ten times and the average residuals are considered for comparison purposes.
>
> --  In [16KLY] and [18JY], fast ergodic convergence rate of DGD was established assuming $T$ is sufficiently large and the step sizes are $\alpha= \frac{1+ \sigma_n}{\rho}$ and $\alpha= \frac{\sigma_n}{\rho}$ for convex and nonconvex objectives, respectively. Our numerical results show efficiency of adaptive algorithms by choosing these parameters. It is worth to mention that recommended $alpha$ for adaptive gradient methods such as ADAM is equal to $0.001$ but this is not optimal for decentralized gradient methods.
>
> [18JY] Zeng Jinshan, and Wotao Yin. "On nonconvex decentralized gradient descent." IEEE Transactions on Signal Processing 66.11 (2018): 2834-2848.
>
> [16KLY]  Yuan, Kun, Qing Ling, and Wotao Yin. "On the convergence of decentralized gradient descent." SIAM Journal on Optimization 26.3 (2016): 1835-1854.
>
> Refer to Appendix on page 29.

---

### Official Review · AnonReviewer3 · 2018-10-28
**Novel algorithm for an important problem but not sufficiently justified theoretically or empirically.**

**Rating:** 4
**Confidence:** 4

**Review:**

This paper presents a consensus-based decentralized version of the Adam algorithm for online optimization. The authors consider an empirical risk minimization objective, which they split into different components, and propose running a separate online optimization algorithm for each component, with a consensus synchronization step that involves taking a linear combination of the parameters from each component before applying each component's individual parameter update. The final output is a simple average of the parameters from each component.

The authors study the important problem of distributed optimization and focus on adapting existing state-of-the-art methods to this setting. The algorithm is clearly presented, and to the best of my knowledge, original. The fact that this work includes both theoretical guarantees for the convex and non-convex settings as well as numerical experiments strengthens the contribution.

On the other hand, I didn't find the actual method presented by the authors to be motivated very well. The main innovation with respect to the standard Adam/AMSGrad algorithm is the use of a mixing matrix W, but the authors do not discuss how the choice of this matrix influences the performance of the algorithm or how one should specify this input in practice. This seems like an important issue, especially since all of the bounds depend on the second singular value of this matrix. Moreover, arguments such as Corollary 10 do not actually imply that DADAM outperforms ADAM when this singular value is large, making it difficult to assess the impact of this work. The numerical experiments also do not test for the statistical significance of the results.

There are also many typos that make the submission seem relatively unpolished.

Specific comments:
1. page 1: "note only". Typo.
2. page 2: "decentalized". Typo.
3. page 2: "\Pi_X[x]. If \Pi_X(x)...." Inconsistent notation.
4. page 3: "largest singular of matrix". Typo.
5. page 3: "x_t* = arg min_{x \in X} f_t(x)". f_t isn't defined in up to this point.
6. page 4: "network cost is then given by f_t(x) = \frac{1}{n} \sum_{i=1}^n f_{i,t}(x)" Should the cost be  \frac{1}{n} \sum_{i=1}^n f_{i,t}(x_{i,t})? That would be more consistent with the definition of regret presented in Reg_T^C.
7. page 4: "assdessed". Typo.
8. page 4: " Reg_T^C := \frac{1}{n} \sum_{i=1}^n \sum)_{t=1}^T f_t(x_{i,t})..." Why is this f_t and not f_{i,t}?
9. page 4: "\hat{v}_{i,t} = v_3 ..." You should reference how this assignment in the algorithm relates to the AMSGrad algorithm. Moreover, you should explain why you chose to use a convex combination in the assignment instead of just the max.
10. page 5: Definition 1. This calculation should be derived and presented somewhere (e.g. in the appendix).
11. page 5: Assumption 3. The notation for the stochastic gradient is not very clear and easily distinguishable from the notation for the deterministic gradient.
12. page 5: Theorem 4. D_T can be very large in the bound, which would make the upper bound meaningless. Can you set hyperparameters in such a way to minimize it? Also, what is the typical size of \sigma_2(W) that one would incur?
13. page 6: Remark 6. This remark seems misleading. It ignores the log(T) and D_T terms, both of which may dominate the data dependent arguments.
14. page 6: "The update rules \tilde{v}_{i,t}...". \tilde{v}_{i,t} is introduced but never defined.
15. page 6: Last display equation. The first inequality seems like it can be an equality.
16. page 7: Equation (14). Doesn't the presence of \sigma_2(W) imply that the O(1/T) term may not be negligible? It would also be helpful to give some examples of how large T needs to be in (15a) and (15b) in order for this statement to take effect.
17. page8: "distributed federated averaging SGD (FedAvg)". What is the reference for this? It should be included here. It should probably also be mentioned in the introduction as related work.
18. page 9: Figure 1. Without error bars, it is impossible to tell the statistical significance of these results. Moreover, how sensitive are these results to different choices of hyperparameters?
19. page 9: "obtain p coefficients". What is p in these experiments?
20. page 9: Metropolis constant edge weight matrix W". What is \sigma_2(W) in this case?
21. page 10: Acknowledgements. This shouldn't be included in the submission.

---

> ### Author Response · Authors · 2018-11-17
> **Response to Reviewer~3**
>
> 1-4 [Comment]
>
> --1. page 1: "note only". Typo.
>
> --2. page 2: "decentalized". Typo.
> .
> .
> .
> --21. page 10: Acknowledgements. This shouldn't be included in the submission.
>
> 1-4 [Response]
>
> Modified and fixed.
>
> 1-5 [Comment]
>
> 9. page 4: "$\hat{v}_{i,t} = v_3 ...$" You should reference how this assignment in the algorithm relates to the AMSGrad algorithm. Moreover, you should explain why you chose to use a convex combination in the assignment instead of just the max.
>
> 1-5 [Response]
>
> -- In some cases, the numerical performance of our experiments is  dependent on the choice of parameter $\hat{v}_{i,t}$ and the results provided establish the efficiency of ADAM in comparison to AMSGrad. Indeed, a good step size value generated in any iterate of ADAM is essentially discarded due to the max in AMSGrad. $\hat{v}_{i,t}$ provides a combination of these two approaches and enables us to develop a convergent adaptive method similar to AMSGrad, while maintaining the efficiency of ADAM.
>
> Refer to Subsection 2 on page 4.
>
> 1-6 [Comment]
>
> 12. page 5: Theorem 4. $D_T$ can be very large in the bound, which would make the upper bound meaningless. Can you set hyperparameters in such a way to minimize it? Also, what is the typical size of $\sigma_2(W)$ that one would incur?
>
> 13. page 6: Remark 6. This remark seems misleading. It ignores the $\log(T)$ and $D_T$ terms, both of which may dominate the data dependent arguments.
>
> 1-6 [Response]
>
> --It is easy to show that the regret of DADAM is bounded by $O(G_\infty D_T \sqrt{T})$ where $D_T =\max_{d \in \{1,...,p\} } D_{T,d}$. Indeed, the term $\sum_{t=1}^T |g_{i,t,d}|/\sqrt{t}$ in the proof of Lemma~13 can be bounded by $O(G_\infty \sqrt{T})$ instead of $O(G_\infty \sqrt{T\log{T}})$. Hence, the regret of DADAM is upper bounded by minimum of $O(G_\infty D_T \sqrt{T})$ and the bound presented in Theorems~4 and 5, and thus the worst case dependence on $T$ is $\sqrt{T}$ rather than $\sqrt{T \log{T}}$. It is worth mentioning that in the static setting, i.e. $D_T=0$, the regret of DADAM is upper bounded by $O(G_\infty \sqrt{T})$.
>
> Refer to Remark 6 on page 6.
>
> 1-7 [Comment]
>
> 16. page 7: Equation (14). Doesn't the presence of $\sigma_2(W)$ imply that the $O(1/T)$ term may not be negligible? It would also be helpful to give some examples of how large T needs to be in (15a) and (15b) in order for this statement to take effect.
>
> 1-7 [Response]
>
> Please see Response 1-3.
>
>
> 1-8 [Comment]
>
> 18. page 9: Figure 1. Without error bars, it is impossible to tell the statistical significance of these results. Moreover, how sensitive are these results to different choices of hyperparameters?
>
>
> 1-8 [Response]
>
> -- The numerical results shown in Figure~1 are based on the deterministic variants of DADAM, DGD and EXTRA algorithms with only local computation and neighbor communication. Indeed, our goal is to show the exact convergence to the reference logistic classifier $\theta^*$. Hence, error bars are provided based on the residual $\|\frac{\theta_T- \theta^*}{\theta - \theta^*}\|$. We have provided a detailed implementation for different choices of hyperparameters in Appendix.
>
> Refer to Figure~1 on page 10.
>
> Refer to Figures~3 and 4 on page 29.

---

> > ### Comment · AnonReviewer3 · 2018-11-22
> > **Comments in response to author response**
> >
> > Thank you for taking the time to respond to my comments as well as for the revisions to the paper.
> >
> > Here are some further comments in response to each of your responses.
> >
> > 1-1) When I said that the actual method wasn't motivated very well, I was referring mostly to the idea of using the mixing matrix W and how it should be specified in practice.
> >
> > On a separate note, I wasn't aware of [DAW12], and I'm surprised that it's not referenced in your paper. In general, I think it would be helper to the reader if you made it more clear what the contribution of your paper is with respect to existing work on decentralized optimization algorithms over networks. For instance, why aren't [DAW12], [07XBK], and [04XB] discussed in the introduction or in Section 1.1? You also spend a lot of time trying to compare the bounds to centralized adaptive methods but provide any discussion on your work in relation to decentralized non-adaptive methods.
> >
> > 1-2) Restricting the mixing matrix W to be symmetric and doubly stochastic still leaves one with a very large family of choices. It's fine to say that optimizing for W isn't the main focus of this work, but its specification is crucial for the performance of the algorithm and its guarantees, so it is important to specify certain choices (which you do with the Metropolis constant edge weight matrix), motivate them (which you don't do), as well as clearly describe their impact on the algorithm's performance (which you also don't do).
> >
> > 1-3) Saying that \sigma_2(W) is strictly less than one is not enough, because it still leaves room for 1- \sigma_2(W) to be arbitrarily small, which can make the bounds arbitrarily large (and therefore meaningless).
> >
> > I inspected the revised bound, and it seems more like a restatement than an improvement. In particular, saying that 1-\sigma_2(W) doesn't appear in the regret bound if T is sufficiently large is misleading, because this can require T to be arbitrarily large.
> >
> > 1-4) That's good.
> >
> > 1-5) It's good that this is now included in the revised version.
> >
> > 1-6) It's not immediately clear to me why that term can be bounded without the \log(T) term. If this can be done, why not just present the improved result?
> >
> > I don't think adding the statement about the static setting at the end of the remark as is done now is very helpful to the reader. If you want to say that DADAM is better in certain specific settings (e.g. static), then you should restate the entire remark more precisely.
> >
> > 1-8) I still don't think I see any error bars in Figure 1. I do see the discussion on hyperparameters, which I think will be helpful to the readers.

---

> > > ### Author Response · Authors · 2018-12-20
> > > **Response to Reviewer~3**
> > >
> > > Again, thank you for your valuable feedback.
> > >
> > >
> > > Comments 1-1, 1-2 and 1-3) [Design of mixing matrix $W$]
> > >
> > > There are several designs for network matrix $W$ in [BPX04],  [TLR12] and [SLWY15]. In earlier papers [TLR12,N015], the role of network constraints on the consensus-based distributed optimization has been analyzed. They provided a unified view of how the network affects both the speed of convergence as well as the solution to which the algorithm converge. However, to the best of our knowledge, there is no general rule for determining the best $W$ in decentralized consensus optimization problems (see, Section IV-B in [TLR12]). We consider the Metropolis constant edge weight matrix [BPX04] here since it is easy to implement and has good performance in general [JXM14,SLWY15].
> > >
> > >
> > > -[BPX04] Boyd, S., Diaconis, P., & Xiao, L. (2004). Fastest mixing Markov chain on a graph. SIAM review, 46(4), 667-689.
> > >
> > > -[TLR12] Tsianos, K. I., Lawlor, S., & Rabbat, M. G. (2012, October). Consensus-based distributed optimization: Practical issues and applications in large-scale machine learning. In Communication, Control, and Computing (Allerton), 2012 50th Annual Allerton Conference on (pp. 1543-1550). IEEE.
> > >
> > > -[NO15] Nedić, A., & Olshevsky, A. (2015). Distributed optimization over time-varying directed graphs. IEEE Transactions on Automatic Control, 60(3), 601-615.
> > >
> > > -[SLWY15] Shi, W., Ling, Q., Wu, G., & Yin, W. (2015). Extra: An exact first-order algorithm for decentralized consensus optimization. SIAM Journal on Optimization, 25(2), 944-966.

---

> > > > ### Author Response · Authors · 2018-12-20
> > > > **Response to Reviewer~3**
> > > >
> > > >
> > > > [Comment]
> > > > It's not immediately clear to me why that term can be bounded without the \log(T) term. If this can be done, why not just present the improved result?
> > > >
> > > > I don't think adding the statement about the static setting at the end of the remark as is done now is very helpful to the reader. If you want to say that DADAM is better in certain specific settings (e.g. static), then you should restate the entire remark more precisely.
> > > >
> > > >  [Response]
> > > >
> > > > The main benefit in ADAM-type methods comes in terms of data sparsity as shown in Theorem~4.  However, similar to AMSGRAD and ADAM, DADAM also has a regret bounded by $G_{\infty} \sqrt{T}$.
> > > >
> > > > Let $\|g_{i,t}\|_{\infty}  \leq G_{\infty} $. Then, the term $\sum_{t=1}^T |g_{i,t,d}|/\sqrt{t}$ in the proof of Lemma~13 can be bounded as follows:
> > > >
> > > > $$\sum_{t=1}^T |g_{i,t,d}|/\sqrt{t}
> > > > \leq \sum_{t=1}^T G_{\infty} /\sqrt{t}
> > > > \leq  G_{\infty}  \int_{t=1}^{T} 1/\sqrt{t}
> > > > \leq G_{\infty} \sqrt{T}.$$
> > > >
> > > > Thus, the upper-bound on DADAM's regret is a minimum between the one in $O(G_{\infty}  \sqrt{T})$ and the one of Theorem~4.
> > > >
> > > > -- Please refer to Remark~7 on page 6.

---

> ### Author Response · Authors · 2018-11-17
> **Response to Reviewer~3**
>
>
> We appreciate the reviewer's constructive comments and suggestions.
> We have carefully addressed them in the revised version of the
> paper and also focused on improving the presentation of the material.
> A detailed point-by-point response to the reviewer's comments follows.
>
> 1-1 [Comment]
>
> --I didn't find the actual method presented by the authors to be motivated very well.
>
> 1-1 [Response]
>
> --Existing distributed stochastic and adaptive gradient methods for various learning problems, including deep learning, are mostly designed for a network topology with a central node. The main bottleneck of such a topology lies on the communication overload on the central node, since all nodes need to concurrently communicate with it.
> Hence, performance can be significantly degraded when network bandwidth is limited. These considerations motivate us to study an adaptive algorithm for network topologies, where all nodes can only communicate with their neighbors and none of the nodes is designated as ``central". Therefore, the proposed method is suitable for large scale machine learning problems, since it enables both data parallelization and decentralized computation.
>
> -- Further, we show that our proposed adaptive distributed algorithm can be faster than its centralized counterpart such as ADAM, ADAGRAD and RMSProp.
>
> Refer to Subsection 1.1 on page 2.
>
>
> 1-2 [Comment]
>
> -- The main innovation with respect to the standard Adam/AMSGrad algorithm is the use of a mixing matrix $W$, but the authors do not discuss how the choice of this matrix influences the performance of the algorithm or how one should specify this input in practice. This seems like an important issue, especially since all of the bounds depend on the second singular value of this matrix.
>
> 1-2 [Response]
>
> -- We assume that the mixing matrix W is symmetric and doubly stochastic (see, equation~1). As mentioned in the Introduction, we consider Metropolis constant edge weight matrix [04XBK, 07XB] (please see, subsection 1.2). When $\hat{W}$ is chosen according to this scheme, $ W = \frac{I+\hat{W}}{2}$ is found to be very efficient [04XBK]. Also, this doubly stochastic matrix implies uniqueness of $\sigma_1(W) = 1$ and warrants that other singular values of $W$ are strictly less than one in magnitude.
>
> -- It is worth mentioning that the optimization of matrix $W$ and in particular $\sigma_2$ is not the main focus of this work. To the best of our knowledge, our theorems are the first to establish a tight connection between the convergence rate of distributed adaptive methods to the spectral properties of the underlying
> network. In particular, the inverse dependence on the spectral gap $1-\sigma_2(W)$ is quite natural, since it is well-known to determine the rates of mixing in random walks on graphs [DAW12, LP17].
>
> ---- [07XBK] Xiao, Lin, Stephen Boyd, and Seung-Jean Kim. "Distributed average consensus with least-mean-square deviation." Journal of parallel and distributed computing 67.1 (2007): 33-46.
>
> ---- [04XB] Xiao, Lin, and Stephen Boyd. "Fast linear iterations for distributed averaging." Systems and Control Letters 53.1 (2004): 65-78.
>
> --- [DAW12] Duchi, John C., Alekh Agarwal, and Martin J. Wainwright. "Dual averaging for distributed optimization: Convergence analysis and network scaling." IEEE Transactions on Automatic control 57.3 (2012): 592-606.
>
> ----[LP17]Levin, David A., and Yuval Peres. Markov chains and mixing times. Vol. 107. American Mathematical Soc., 2017.
>
> Refer to Subsection~1.2 on page 2.
>
> 1-3 [Comment]
>
> Arguments such as Corollary 10 do not actually imply that DADAM outperforms ADAM when this singular value is large, making it difficult to assess the impact of this work. The numerical experiments also do not test for the statistical significance of the results.
>
> 1-3 [Response]
>
> -- First, the doubly stochastic matrix $W$ defined by our strategy in Section~1.2 warrants that $\sigma_2(W)$ is strictly less than one in magnitude and for a fully connected network is actually equal to 0.
>
> -- Second, in the revised version, we improve the previous result and the spectral gap $1- \sigma_2(W)$ does not appear in the regret bound of DADAM, if $T$ is sufficiently large.
>
> -- Finally, in the context of stochastic nonconvex optimization, we say a gradient-based method gives an $\epsilon$-approximation solution if $T^{-1}\e{{\bf Reg}^N_T} \leq \epsilon,$ where ${\bf Reg}^N_T$ is defined in Section~2.
> Now, assume that $T$ is sufficiently large, i.e. it satisfies (16a) and (16b), the $\frac{1}{T}$ term will be dominated by the $\frac{1}{\sqrt{nT}}$ term which leads to a $\frac{1}{\sqrt{nT}}$ convergence rate where $n$ is the number of agents. More specifically, it shows that the computation complexity of DADAM to achieve $\epsilon$-approximation solution is  $O(1/\epsilon^2)$. This shows that DADAM can be faster than ADAM for nonconvex stochastic optimization problems for $T$ sufficiently large.
>
>  Refer to Subsection 3.2.1.

---

### Official Review · AnonReviewer1 · 2018-11-05
**This paper proposes a consensus-based distributed method, namely DADAM, for online optimization. The technical details are well presented and the empirical results are convincing.**

**Rating:** 8
**Confidence:** 3

**Review:**

The proposed DADAM is a sophisticated combination of decentralized optimization and the adaptive moment estimation. DADAM enables data parallelization as well as decentralized computation, hence suitable for large scale machine learning problems.

Corollary 10 shows better performance of DADAM. Besides the detailed derivations, can the authors intuitively explain the key setup which leads to this better performance?

The experimental results are mainly based on sigmoid loss with simple constraints. The results will be more convincing if the authors can provide studies on more complex objective, for example, regularized loss with both L2 and L1 bounded constraints.

Th experimental results in Section 5.1 is based on \beta_1 = \beta_2 = \beta_3 = 0.9. From  the expression of \hat v_{i,t} in Section 2, this setting implies the most recent v_{i,t} plays a more important role than the historical maximum, hence ADAM is better than AMSGrad. I am curious what the results will look like if we set \beta_3 as a value smaller than 0.5.

---

> ### Author Response · Authors · 2018-11-27
> **Response to Reviewer~1**
>
> We thank the reviewer for the helpful and supportive feedback. A detailed point-by-point response to the reviewer's comments follows.
>
> 1-1 [Comment]
>
> -- Corollary 10 shows better performance of DADAM. Besides the detailed derivations, can the authors intuitively explain the key setup which leads to this better performance?
>
> 1-1 [Response]
>
> The key setup which leads to this regret bound is that we do not use the boundedness assumption for domain or gradient. These assumptions may simplify the proof but lose some sophisticated structures in the distributed optimization problems.  Further, the advantage of DADAM over centralized parallel gradient methods is to avoid the communication traffic jam. More specifically, the communication cost for each node of DADAM is O(the degree of the graph)  which could be much smaller than $O(n)$ for centralized gradient-based methods .
>
>
> Refer to Paragraph~2 on page 8.
>
>
> 1-2 [Comment]
>
> -- The experimental results are mainly based on sigmoid loss with simple constraints. The results will be more convincing if the authors can provide studies on more complex objective, for example, regularized loss with both L2 and L1 bounded constraints.
>
> 1-2 [Response]
>
> --  We have provided a detailed implementation for different choices of regularized loss with both L2 and L1 bounded constraints.
>
> Refer to Equation ~18 on page 9.
> Refer to Figure~1 on page 10.
>
>
> 1-3 [Comment]
>
> -- Th experimental results in Section 5.1 is based on \beta_1 = \beta_2 = \beta_3 = 0.9. From  the expression of \hat v_{i,t} in Section 2, this setting implies the most recent v_{i,t} plays a more important role than the historical maximum, hence ADAM is better than AMSGrad. I am curious what the results will look like if we set \beta_3 as a value smaller than 0.5.
>
> 1-3 [Response]
>
> -- In Appendix, we examine the sensitivity of DADAM on the parameters related to the network connection and update of the moment estimate. We consider a range of hyperparameter choices, i.e. $\beta_3 \in {0,0.9,0.99}$. From Figure 4 it can be easily seen that DADAM performs equal or better than AMSGrad $(\beta_3 = 0)$, regardless of the hyper-parameter setting for  $\beta_1$ and $ \beta_2$.
>
> Refer to Figures~3 and 4 on page 29.

---

### Public Comment · (anonymous) · 2018-11-18
**An Interesting Optimization Problem**

Dear Authors:
I appreciate the interesting work authors present in this paper. One question is about the convergence of DADAM on the nonconvex case. Can DADAM converge to a critical point? Thank you.

---

> ### Author Response · Authors · 2018-11-18
> **Convergence Rate of DADAM for Non-Convex Objectives**
>
> Thank you for your interest in the paper.
>
> Let $f$ be real-valued, continuously differentiable (possibly nonconvex) function on a closed, convex set $\mathcal{X}$. The projected gradient $G_{\mathcal{X}}(x,f,\alpha)$ can be used to characterize stationary points because if $\mathcal{X}$ is a convex set, then $ x \in \mathcal{X}$ is a stationary point or critical point of continuously differentiable function $f$ if and only if $G_{\mathcal{X}}(x,f,\alpha) =0 $ [87CM, ABS13]. In general, $G_{\mathcal{X}}(x,f,\alpha)$ is discontinuous, but as proved by Calamai and More [87CM], if $f$ is continuously differentiable on $\mathcal{X}$, then the mapping $x \rightarrow \|G_{\mathcal{X}}(x,f,\alpha)\|$ is lower semicontinuous on $\mathcal{X}$. This property implies that if ${x_t}$ is a sequence in $\mathcal{X}$ that converges to $x^*$, and if $G_{\mathcal{X}}(x_t,f,\alpha)$ converges to zero, then $x^*$ is a stationary point of problem \ref{125}. Motivated by [87CM,ABS13,HSZ17], we monitor convergence to a stationary point using the \textit{local regret} which is an extension of projected gradients to the online distributed settings (please see Definition~1).
>
> --In Theorem~7, we analyze the convergence of DADAM for general Lipschitz and smooth (possibly non-convex) loss function using the local regret and show that the online distributed algorithms converge even when the loss is non-convex, i.e., the algorithms find a stationary point to the time-varying loss at a rate of $\tilde{O}(\frac{1}{T})$.
>
> --In Theorem 9, we extend this result to the stochastic nonconvex settings when noisy gradients are accessible to the agents. Finally, in Corollary~10, we show the potential advantage of DADAM over adaptive algorithms such as ADAM, ADAGRAD and RMSProp for solving stochastic nonconvex optimization problems. More specifically, our theoretical results show that DADAM can be faster than adaptive algorithms for finding stationary points of stochastic non convex problems when $T$ is sufficiently large.
>
>
> ---- [87CM] Calamai, Paul H., and Jorge J. Moré. "Projected gradient methods for linearly constrained problems." Mathematical programming 39.1 (1987): 93-116.
>
> ----  [ABS13] Attouch, H., Bolte, J., & Svaiter, B. F. (2013). Convergence of descent methods for semi-algebraic and tame problems: proximal algorithms, forward–backward splitting, and regularized Gauss–Seidel methods. Mathematical Programming, 137(1-2), 91-129.
>
> ---- [HSZ17] Hazan, Elad, Karan Singh, and Cyril Zhang. "Efficient Regret Minimization in Non-Convex Games." arXiv preprint arXiv:1708.00075 (2017).

---

> > ### Public Comment · (anonymous) · 2018-11-21
> > **For a general deep neural network, can DADAM converge to a critical point**
> >
> > Dear Authors:
> >                  Thank you for the explanation. You propose a general online optimization method, but can you prove that it converges to a critical point in a deep neural network problem? Notice that  the objective function may be non-differentiable(like relu). Thank you.
> >                   Sincerely yours

---

> > > ### Author Response · Authors · 2018-11-24
> > > **Convergence Rate of DADAM for Non-Smooth Objectives**
> > >
> > >  Thanks for the interest in our paper and looking into the analysis carefully.
> > >
> > > -- Performance of SGD is best judged by its sample complexity which is related to the regularity of  objective $F(x)$.  For convex objective $F(x)$ the stochastic (sub)-gradient method [C85] attains expected functional accuracy $\epsilon$ with after $O(\epsilon^{-2})$ stochastic sub-gradient evaluations .  However, for non-convex non-smooth problems, the situation is less clear.  The challenge in establishing a sample complexity  for non-smooth non-convex sub-gradient-based methods is that the “ convergence criteria,” namely the objective error $F(x_t) - \inf F$ and the norm of the subgradient can be completely meaningless . Indeed, one cannot expect $F(x_t) - \inf F$ to tend to zero---even in the smooth setting. Also, simple examples, e.g., $F(x) = |x|$, show that $\dist(0, \partial F(x_t))$ can be strictly bounded below by a fixed constant for all iterations.
> > >
> > > -- In contrast to subgradient-based methods, the ``"convergence criteria" is meaningful for the \emph{proximal  sub-gradient methods}~[R76], which constructs $x_{t+1}$ by approximately minimizing the subproblem
> > > $
> > >  \min_{x \in \mathbb{R}^p} \left\{ F(x) + \frac{1}{2c_t}\|x - x_t\|^2\right\},
> > > $
> > > where $c_t$ is a control parameter.  Indeed, it is easy to show that under minimal assumptions on $F$, the subdifferential distance $\dist(0, \partial F(x_{t}))$ tends to zero (please see Theorem~1 in [R76]).
> > >
> > > -- In this paper, we provide the complexity guarantees for an adaptive distributed gradient-based method for a general class of smooth losses in online  and stochastic settings. However, the guarantees in this paper apply to  the non-smooth settings by using a proximal point scheme similar to [R76, DD18] that may be summarized as follows
> > > $
> > > x_{t+1} = \argmin_{x \in \mathbb{R}^p} \left\{ \frac{1}{n}\sum_{t=1}^T\sum_{i=1}^n f_{i,t}(x) + \frac{1}{2c_t}\|x - x_t\|^2\right\} .
> > > $
> > > where $c_t$ is a control parameter.
> > >
> > > ---- [C85] Blair, Charles. "Problem complexity and method efficiency in optimization (as nemirovsky and db yudin)." SIAM Review 27.2 (1985): 264.
> > >
> > > ---- [R76] Rockafellar, R. Tyrrell. "Monotone operators and the proximal point algorithm." SIAM journal on control and optimization 14.5 (1976): 877-898.
> > >
> > > ---- [ DD18] Davis, Damek, and Dmitriy Drusvyatskiy. "Stochastic model-based minimization of weakly convex functions." arXiv preprint arXiv:1803.06523 (2018).

---

> > > > ### Public Comment · (anonymous) · 2018-11-30
> > > > **Thank you for the explanation**
> > > >
> > > > Dear Authors:
> > > >                Thank you for the explanation. I need to check the proof carefully. Anyway, this is an interesting paper and hope you can get accepted.
> > > >                   Sincerely yours

---

### Author Response · Authors · 2019-01-29
**ICLR 2018 Conference Acceptance Decision**


We have taken the feedback seriously and improved the paper substantially; see  https://arxiv.org/pdf/1901.09109.pdf

The employed data sets and software code are available at:  https://github.com/Tarzanagh/DADAM

---

### Meta-Review · Area_Chair1 · 2018-12-18
**Borderline paper: distributed optimization algorithm with analysis**

**Confidence:** 4
**Recommendation:** Reject

**Metareview:**

The paper provides a distributed optimization method, applicable to decentralized computation while retaining provable guarantees.  This was a borderline paper and a difficult decision.

The proposed algorithm is straightforward (a compliment), showing how adaptive optimization algorithms can still be coordinated in a distributed fashion.  The theoretical analysis is interesting, but additional assumptions about the mixing are needed to reach clear conclusions: for example, additional assumptions are required to demonstrate potential advantages over non-distributed adaptive optimization algorithms.

The initial version of the paper was unfortunately sloppy, with numerous typographical errors.  More importantly, some key relevant literature was not cited:
- Duchi, John C., Alekh Agarwal, and Martin J. Wainwright. "Dual averaging for distributed optimization: Convergence analysis and network scaling." IEEE Transactions on Automatic control 57.3 (2012): 592-606.
In addition to citing this work, this and the other related works need to be discussed in relation to the proposed approach earlier in the paper, as suggested by Reviewer 3.

There was disagreement between the reviewers in the assessment of this paper.  Generally the dissenting reviewer produced the highest quality assessment.  This paper is on the borderline, however given the criticisms raised it would benefit from additional theoretical strengthening, improved experimental reporting, and better framing with respect to the existing literature.